# Private Learning Fast and Slow: Two Algorithms for Prediction with Expert Advice Under Local Differential Privacy

## Abstract

We study the classic problem of prediction with expert advice under the constraint of differential privacy (DP). In contrast to earlier work in this area, we are interested in distributed settings with no trusted central curator. In this context, we first show that a classical online learning algorithm naturally satisfies DP and then design two new algorithms that extend and improve it: (1) RW-AdaBatch, which provides a novel form of privacy amplification at negligible utility cost, and (2) RW-Meta, which improves utility on non-adversarial data with zero privacy cost. Our theoretical analysis is supported by an empirical evaluation using real-world data reported by hospitals during the COVID-19 pandemic. RW-Meta outperforms the classical baseline at predicting which hospitals will report a high density of COVID-19 cases by a factor of more than $2\times$ at realistic privacy levels.

## 1 Introduction

Many practical algorithmic and organizational problems, from routing data to allocating scarce resources, involve repeated predictions about the future state of a system. Prediction with expert advice (Cesa-Bianchi & Lugosi, 2006) is a flexible framework developed in the online learning literature that can be used to model these problems as an iterative game between a player and nature: at each time step $t = 1, \ldots, T$, the player receives suggestions from each of $n$ 'experts' and must decide whose advice to follow. Nature then reveals the *gain* of each expert at that time step, and the player receives a reward based on their actual choice. The player's goal is to maximize their total reward over the course of the entire game.

Over the past several decades, the field of online learning has developed many sophisticated algorithms for this problem with impressive performance guarantees (Adamskiy et al., 2012; Luo & Schapire, 2015; Korotin et al., 2020). In many applications, however, predictions are either implicitly or explicitly about human behavior. Optimizing purely for predictive accuracy in this context is risky because the algorithm might inadvertently leak sensitive data through its outputs, analogous to the well-understood privacy risks in batch learning (Shokri et al., 2017; Carlini et al., 2023).

To mitigate this issue, researchers have developed several experts algorithms that provably satisfy differential privacy (DP), a popular method for controlling the information that a statistical algorithm leaks about its input (Dwork et al., 2006). But despite strong theoretical results, these algorithms rely on several common assumptions that limit their practical application. Our goal in this work is to overcome this limitation by designing algorithms that rely on more general assumptions, allowing them to be used in contexts where current state-of-the-art methods cannot be easily applied.

### 1.1 Our Contributions

Our main contribution is the design, analysis, and evaluation of two new algorithms for prediction with expert advice under local DP: RW-AdaBatch and RW-Meta. We describe the specific advantages of our approach below.

**More realistic assumptions.** Unlike prior work, our algorithms are compatible with the local model of DP, which does not assume the existence of a trusted central data curator. This makes them applicable to settings like healthcare, transportation, or energy usage where predicting aggregate behavior

is socially useful but individual-level records are often too sensitive to share. Also unlike existing work, our algorithms are *anytime* (Dean & Boddy, 1988), meaning that they do not require advanced knowledge of the input size to optimize parameters.

**Novel form of privacy amplification.** The RW-AdaBatch algorithm offers amplified privacy guarantees by adaptively batching incoming data points together as they arrive. This gives each of the batched points a "crowd" to hide in, improving the final privacy analysis of the algorithm's outputs in a manner analogous to the celebrated shuffle model (Erlingsson et al., 2019). Our analysis leverages the theory of random walks to prove that this batching carries zero utility cost with high probability.

**Private learning in dynamic environments.** The RW-Meta algorithm leverages meta-learning to greatly improve performance in dynamic environments at no additional privacy cost. The key challenge is that in the experts problem, it is possible to leak information through both our choice of expert and the actual advice the chosen expert gives us. Prior work addresses this issue by implicitly assuming that expert advice is independent of any sensitive data, but this is a severe limitation. To overcome this, we exploit correlated noise and linearity to simultaneously compute advice for many data-aware experts *and* the optimal choice of expert, all at a fixed privacy cost. To the best of our knowledge, the resulting private experts algorithm is the first that is capable of achieving high absolute performance on inputs where the best action varies dynamically over time.

**Empirical validation.** We rigorously evaluate our algorithms' performance using a combination of synthetic data and real-world data reported by hospitals during the COVID-19 pandemic. On the privacy side, we show that the the privacy amplification of RW-AdaBatch is substantial and only grows over time, corresponding e.g. to a nearly $10\times$ improvement in the worst-case value of the privacy parameter $\delta$ when $t = 10,000$. On the utility side, we find that RW-Meta outperforms the classical algorithm by a factor of more than $2\times$ at predicting which hospitals will report the highest density of COVID-19 patients each week. It would be essentially impossible for any algorithm based on data-independent experts to match the performance we observe.

## 2 RELATED WORK

The study of online learning with DP was initiated in 2010 by the papers of Dwork et al. (2010) and Chan et al. (2011), which examined the problem of privately updating a single counter. Later work has gone on to apply their framework to more complicated statistics (Perrier et al., 2018; Bolot et al., 2013; Wang et al., 2021). Jain et al. (2012) formalized the idea of private online learning through the lens of convex optimization and designed algorithms for optimizing strongly convex functions while preserving privacy. Subsequently, Thakurta & Smith (2013) extended the study of private online learning to include the partial information or bandit setting, where the learner only observes the reward for the specific action they chose, which has also been studied in Hannun et al. (2019) and Azize & Basu (2024). Recently, considerable interest in private online convex optimization has been driven by machine learning where it is used to study the properties of DP-SGD and related algorithms (Kairouz et al., 2021; Choquette-Choo et al., 2023; 2024).

The works studying private prediction with expert advice that are most similar to ours are Asi et al. (2023) and Agarwal & Singh (2017), which both fall under the central model of DP. We also note that very recent parallel work by Gao et al. (2024) explores online prediction from experts in the federated setting, where $m$ clients all attempt to solve a single experts problem simultaneously with limited communication. Their Fed-DP-OPE-Stoch algorithm is the first experts algorithm that we are aware to explicitly satisfy local DP, and therefore shares motivation with our work. However, the federated version of the problem and the stochastic adversaries they consider lead to substantial technical differences between our results.

Like us, Asi et al. (2023) exploit the infrequent-switching behavior of a classical online learning algorithm (Shrinking Dartboard, by Geulen et al. (2010)) to develop differentially private algorithms for prediction from expert advice. Their focus is on the high-dimensional regime, where the gap between local and central DP is most pronounced (Edmonds et al., 2019; Duchi et al., 2013).

The algorithms in Agarwal & Singh (2017) represent the state-of-the-art in the low-dimensional regime that we target. All of the algorithms developed in that work rely on the binary tree aggregation technique (Dwork et al., 2010; Chan et al., 2011), which can be used to simultaneously estimate

any partial sum of a sequence with $O(\log^{1.5} T)$ noise scale while satisfying DP. While highly efficient, this technique fundamentally requires the time horizon $T$ to be known in advance, and also assumes the existence of a central trusted data curator.

Our work overcomes these limitations at the cost of a weaker worst-case dependence on the dimension of the problem, which is a consequence of satisfying local vs. central DP (Duchi et al., 2013). We therefore emphasize that our goal is *not* to directly compete with these state-of-the-art algorithms in the settings they target (i.e. central DP, static data, known time horizon). Rather, we aim to design algorithms with weaker starting assumptions that can perform well in settings that the SOTA cannot be (easily) applied to (i.e. local DP, shifting data distributions, unknown time horizon).

## 3 PROBLEM SETTING

### 3.1 NOTATION

Given a vector $v \in \mathbb{R}^n$, we denote its $k^{th}$ element with the subscript $v_k$, and its $k^{th}$ *smallest* element using parentheses as in $v_{(k)}$. We use the term 'leader' to refer to the argmax over the elements of a vector $v$, and the term 'gap' to refer to the quantity $v_{(n)} - v_{(n-1)}$. For two random variables $X$ and $Y$, we say that $X$ is stochastically larger than $Y$, denoted $X \geq_{st} Y$, if for all $x$ we have $\mathbb{P}[X > x] \geq \mathbb{P}[Y > x]$. We use $\Phi$ and $\varphi$ to denote the standard Gaussian CDF and PDF respectively.

### 3.2 PREDICTION WITH EXPERT ADVICE

Prediction with Expert Advice can be thought of as a special case of online linear optimization (Abernethy et al., 2014). At each time step $t \in [T]$, we choose an action $x_t$ from the action set $\mathcal{X}$, which in our case is the $n$-dimensional probability simplex. We then observe the gain vector $g_t \in [0, 1]^n$ and receive reward $\langle x_t, g_t \rangle$. A typical goal is to minimize *static regret*, which is defined as the difference between the reward we actually receive and the reward of the best single action in hindsight, i.e. $\max_{x \in \mathcal{X}} \sum_{t=1}^{T} \langle x, g_t \rangle - \sum_{t=1}^{T} \langle x_t, g_t \rangle$.

Many algorithms are known which can achieve optimal static regret bounds of $O(\sqrt{T \log n})$ (Cesa-Bianchi & Lugosi, 2006). Unlike statistical learning, these guarantees do not rest on any distributional assumptions and still hold even when the data is adversarially generated. In the *oblivious adversary* model, the gain vectors are fixed in advance (with full knowledge of the algorithm), while in the *adaptive adversary* model, incoming gain vectors are allowed to depend on our earlier decisions. These models turn out to be equivalent in the non-private setting, but recent work has shown that very private algorithms can be forced to incur $O(T)$ regret by adaptive adversaries (Asi et al., 2023). We focus exclusively on oblivious adversaries in this work.

### 3.3 STATIC VS. DYNAMIC ENVIRONMENTS

The definition of static regret makes the most sense in contexts where sequential data points are roughly independent, e.g. because they are drawn i.i.d. from some fixed distribution or because they are truly adversarial. In practice, however, it is often the case that the best action can vary systematically over time, in which case tight bounds on static regret do not imply strong absolute performance. To address this issue, the online learning community has developed many algorithms that achieve tight bounds with respect to stronger notions of regret (Zinkevich, 2003; Hazan & Seshadhri, 2009; Herbster & Warmuth, 1998), but no comparable line of work yet exists in the private online learning literature. We will design (private) algorithms for both settings.

### 3.4 LOCAL DIFFERENTIAL PRIVACY

We consider a distributed setting with a single server and multiple clients $c \in \mathcal{C}$. Each client represents a set of users which can potentially vary over time; we represent the data held by client $c$ at time $t$ as $D_{c,t}$. The goal of the analyst controlling the server is to solve an experts problem where gain is defined in terms of local statistics of these datasets, i.e. $g_t = f(g(D_{1,t}), \ldots, g(D_{|\mathcal{C}|,t}))$. We assume that clients are mutually untrusting of one another as well as the server, however, and are not

willing to share their data directly. To resolve this tension, we would like to devise a method whereby each client can share only a noisy approximation $\tilde{g}(D_{c,t}) \approx g(D_{c,t})$, simultaneously protecting the privacy of the individual data subject(s) and enabling the server to make accurate predictions.

Our results are mostly agnostic to the exact way that gain vectors are defined in terms of the clients — the only requirement is that at each time step, the server receives (information equivalent to) a Gaussian approximation of the true gain vector $\tilde{g}_t \sim \mathcal{N}(g_t, \eta^2 I_n)$. In the simplest case, only one client $c_t$ reports data at each time step and the server receives $\tilde{g}(D_{c_t,t}) = g(D_{c_t,t}) + z$, where $z \sim \mathcal{N}(0, \eta^2 I_n)$. In section 5, we also consider a more complicated setting where each client computes a single noisy *coordinate* of $g_t$ and the server concatenates their reports. We focus on the non-interactive setting where clients send a single report per time step and do not collaborate.

To formally define the privacy property we want to satisfy, we use the popular framework of local differential privacy (LDP) (Dwork et al., 2014; Yang et al., 2023). At a given time step, we say that two sequences of databases are adjacent, denoted $\{D_{c,t}\} \simeq \{D'_{c,t}\}$, if they differ only in one individual's data — the exact meaning of this depends on contextual details, like whether it makes sense for individuals to switch between clients. In the local model of DP that we use, the goal is to make it difficult for the analyst to distinguish adjacent sequences given the clients' reports. In the weaker central model of DP, the analyst is instead given unrestricted access to the data and the goal is to make it difficult to distinguish adjacent sequences given our algorithm's *outputs*.

Most of our technical results are expressed in the language of $f$-DP (Dong et al., 2019), a modern DP variant based on hypothesis testing. Given two distributions $P, Q$, we define the tradeoff function $\mathcal{T}(P, Q) : [0, 1] \rightarrow [0, 1]$ so that $T(P, Q)(\alpha)$ is the minimum false negative rate achievable at false positive rate $\alpha$ when distinguishing $P$ from $Q$. With slight abuse of notation, we say that our protocol satisfies event-level, local $f$-DP if $T((\tilde{g}(D_{1,t}), \ldots, \tilde{g}(D_{|C|,t})), (\tilde{g}(D'_{1,t}), \ldots, \tilde{g}(D'_{|C|,t}))) \geq f$ for all pairs of adjacent sequences. As a special case, we define the Gaussian tradeoff functions $G_\mu := \mathcal{T}(\mathcal{N}(0, 1), \mathcal{N}(\mu, 1))$ and say that a mechanism satisfies $\mu$-Gaussian DP ($\mu$-GDP) if it satisfies $G_\mu$-DP. This definition can be naturally satisfied by locally injecting Gaussian noise with scale calibrated to the sensitivity of our gain vectors, defined as $\Delta = \max_{\{D\} \simeq \{D'\}} \|g_t - g'_t\|$.

To facilitate comparison of our privacy guarantees with earlier work, we will also make some use of the older, more traditional definition of approximate DP: we say that our protocol satisfies (event-level, local) $(\varepsilon, \delta)$-DP if for all adjacent sequences and all measureable events $S$, we have that $\mathbb{P}[(\tilde{g}(D_{1,t}), \ldots, \tilde{g}(D_{|\mathcal{C}|,t})) \in S] \leq e^\varepsilon \mathbb{P}[(\tilde{g}(D'_{1,t}), \ldots, \tilde{g}(D'_{|\mathcal{C}|,t})) \in S] + \delta$.

## 4 PREDICTION WITH EXPERT ADVICE UNDER LOCAL DP

Our goal is to design experts algorithms for both static and dynamic environments which satisfy local differential privacy. We will begin by presenting the main ideas behind our results for static settings — full proofs of all results can be found in the appendix.

### 4.1 CLASSICAL ALGORITHM

The classic algorithm that inspired our approach was introduced in Devroye et al. (2013) under the name 'Prediction by random-walk perturbations.' We will refer to it as RW-FTPL for brevity. Given a symmetric distribution $\mathcal{D}$, the algorithm first samples random variables $z_0, \ldots, z_T \overset{iid}{\sim} \mathcal{D}$. At each time step, it chooses $x_t = \arg\max_{i \in [n]} (G_{t-1} + S_{t-1})_i$, where $S_t := \sum_{s=0}^{t} z_t$ is an element of a symmetric random walk and $G_t = \sum_{s=0}^{t} g_s$. We are interested in the case where $\mathcal{D} = \mathcal{N}(0, \eta^2 I_n)$ and the $S_t \sim \mathcal{N}(0, (t+1)\eta^2 I_n)$ are elements of a symmetric Gaussian random walk.

Using tools from convex analysis, it can be shown that this simple algorithm has expected static regret at most $\left(\eta + \frac{2}{\eta}\right)\sqrt{2T \log n}$ (see e.g. Section 3.5.2 of Lee (2018)). These bounds are optimal up to a constant factor when $\eta$ is independent of the dimension. In the worst case, they can be weaker

---

**Algorithm 1** RW-AdaBatch

---

**Require:** Noise scale $\eta$, dimension $n$, tolerance $\alpha$

$\tilde{G} \sim \mathcal{N}(0, \eta^2 I_n)$

Set initial delay to 0

**for** $t = 1, \ldots, T$ **do**

    Server chooses $x_t = \arg\max_{x \in \mathcal{X}} \langle x, \ \tilde{G} \rangle$

    Clients send noisy vectors $\tilde{g}(D_{c,t})$ to server

    Server computes $\tilde{g}_t \sim \mathcal{N}(g_t, \eta^2 I_n)$ from the noisy vectors and adds it to the buffer

    **if** delay is 0 **then**

        Server moves data from buffer to $\tilde{G}$

        Server computes new delay using Theorem 4.1, $\alpha$, $n$, $\eta$ and $\tilde{G}$      $\triangleright$ See subsection A.3

    **end if**

**end for**

---

than the non-private baseline by a factor of $\sqrt{n}$, matching known lower bounds for mean estimation in high dimensions under local DP Duchi et al. (2013).

When analyzing the privacy guarantees of the algorithm, it is helpful to view the decision rule from a different perspective. Define $\tilde{g}_t := g_t + z_t$, with $g_0 = 0$. RW-FTPL can be reformulated as a post-processing of these noisy gain vectors, which by our assumptions about the problem setting satisfy local GDP with privacy parameter $\mu = \Delta / \eta$. It follows by parallel composition (McSherry, 2009) that RW-FTPL as a whole satisfies local GDP with the same value of $\mu$.

## 4.2 STATIC ENVIRONMENTS: ADAPTIVE BATCHING

The RW-FTPL algorithm was not originally designed with privacy in mind. The authors' primary goal was to create a low-regret algorithm which changes its prediction only a small number of times in expectation. Variations on this goal have been explored extensively in the online learning literature (Kalai & Vempala, 2005; Geulen et al., 2010; Altschuler & Talwar, 2018), and Asi et al. (2023) observe that there is a conceptual connection between privacy and limited switching, as both constraints limit the ways data can be used.

In this section, we develop this intuition by connecting the limited switching of RF-FTPL with the idea of permutation-invariance, which is a critical tool in amplifying the central DP guarantees of local DP algorithms (Erlingsson et al., 2019). The high level idea is that in some cases, we can reliably predict that our algorithm will not switch predictions in the near future no matter what data it sees. When this happens, we can safely batch together all incoming data for some time before updating the algorithm's internal state with all of the points simultaneously. This makes the algorithm fully invariant to permutations within each batch, amplifying its central DP guarantee.

**Utility Analysis.** We show that the expected regret of RW-AdaBatch is at most $(1 + \sqrt{2}\alpha)$ times greater than the expected regret of RW-FTPL, where $\alpha$ is a small constant given to the algorithm as a parameter. Our main tool for proving this result is the following, which establishes the conditions under which RW-FTPL is very unlikely to change its prediction in the near future:

**Theorem 4.1.** *Let $x_0, x_1, \ldots, x_B \in \mathbb{R}^n$ be a Gaussian random walk with $x_0 = v$ and $x_{t+1} - x_t \sim \mathcal{N}(0, \eta^2 I_n)$. If $v_{(n)} - v_{(n-1)} = k$, then the probability that the leader changes at any point during the random walk is at most $2\Phi(-\sqrt{2}\beta) + 2\sqrt{\pi}\varphi(-\beta)\big[\Phi(\beta) - \Phi(-\beta)\big]$, where $\beta = k/(\eta\sqrt{2B}) - \sqrt{\log(2n-2)}$. The same is true if $v_{(n)} - v_{(n-1)} = k + x$ and we wish to bound the probability that the gap ever dips below $x$.*

To see how this helps us, suppose that we are at time step $t$ and must decide how many new points to batch together. RW-AdaBatch works by computing the largest batch size $B_t$ such that the probability of RW-FTPL changing its prediction in the next $B_t$ time-steps is at most $\delta_t = \alpha\sqrt{\log n/(t + B_t)}$. This guarantees that the extra expected regret incurred by RW-AdaBatch at any time step $\tau$ is at most $\alpha\sqrt{\log n/\tau}$, and so the extra expected regret over the entire input is at most $2\alpha\sqrt{T \log n}$.

**Privacy Analysis.** We show that RW-AdaBatch offers a stronger central DP guarantee than RW-FTPL at a fixed local privacy level. This is because it is designed to be fully invariant to permutations of data points within a single batch, which is a fundamental tool for amplifying the central DP guaranteed of locally private algorithms (Erlingsson et al., 2019). Although the exact degree of amplification lacks a closed form, we derive numerically-computable lower bounds on the resulting tradeoff function and empirically validate their tightness in section 5. We note that it is also possible to amplify the privacy guarantee against the *analyst* to match the central DP guarantee if we have access to a trusted third party to shuffle each batch before sending it to the server.

With or without a trusted third party, quantifying the exact degree of amplification requires characterizing the distribution of the size of the batch containing a given individual's data. The main challenge here is that subsequent batch sizes are neither independent nor identically distributed. We circumvent this issue by reducing to the simpler problem of characterizing the distribution of the gap size at a given time step, regardless of whether that time step is the first in a new batch:

At each time step $t$, there is a minimum threshold $k_{t,B}^*$ such that our algorithm will select a batch size of at least $B+1$ at time $t$ iff the gap $k_t \geq k_{t,B}^*$. For a fixed $B$, this threshold is monotonically increasing in $t$ because $\delta_t$ is monotonically decreasing. So, the time $t$ can fall in a batch of size $B+1$ or less only if there exists a time step $s \in [t-B, \ldots, t]$ such that $k_s < k_{t,B}^*$. Therefore, it suffices to first characterize the the distribution of $k_{t-B}$, and then use Theorem 4.1 to bound the probability that the gap dips below $k_{t,B}^*$ at any point prior to time step $t$ given that it starts out larger.

The following theorem shows that worst case distribution of $k_{t-B}$ occurs when the gains of all experts are equal:

**Theorem 4.2.** *Let $S_\varepsilon = \{v \in \mathbb{R}^n : v_{(n)} - v_{(n-1)} \leq \varepsilon\}$ and let $\gamma$ denote the standard Gaussian measure on $\mathbb{R}$. Then for any vector $\mu \in \mathbb{R}^n$, $\gamma^n(S_\varepsilon) \geq \gamma^n(S_\varepsilon - \mu)$*

Theorem 4.2 enables us to reduce all questions about the size of the gap at a given time step to the distribution of the gap of a unit variance multivariate Gaussian random variable, with CDF and PDF:

$$F_k(\varepsilon) = 1 - n \int_{-\infty}^{\infty} \varphi(x)\Phi(x-\varepsilon)^{n-1} \, dx \tag{1}$$

$$f_k(\varepsilon) = n(n-1) \int_{-\infty}^{\infty} \varphi(x)\varphi(x-\varepsilon)\Phi(x-\varepsilon)^{n-2} \, dx \tag{2}$$

Additionally, we have the following lemma which allows us to translate an upper bound on the CDF of batch sizes into a lower bound on tradeoff functions:

**Lemma 4.2.1.** *Let $s_1 \sim P$ and $s_2 \sim Q$ for some distributions $P \geq_{st} Q$ over the non-negative real numbers. Then $\mathcal{T}\big((s_1, \mathcal{N}(0, s_1^2), (s_1, \mathcal{N}(1, s_1^2))\big) \geq \mathcal{T}\big((s_2, \mathcal{N}(0, s_2^2), (s_2, \mathcal{N}(1, s_2^2))\big)$.*

We can therefore bound the probability that a point falls in a batch of size $B+1$ or less by integrating the upper bound from Theorem 4.1 and translating the resulting bound on the CDF into a bound on the actual tradeoff function. All that remains is to directly evaluate this tradeoff function, for which we require the following lemma from Wang et al. (2024):

**Lemma 4.2.2** (Joint Concavity of Tradeoff Functions)**.** *Let $P_w, Q_w$ be two mixture distributions, each with $m$ components and shared weights $w$. Then: $\mathcal{T}(P_w, Q_w)(\alpha(t,c)) \geq \sum_{b=1}^m w_b \mathcal{T}(P_b, Q_b)(\alpha_b(t,c)) =: \beta(t,c)$, where $\alpha_b(t,c) = \mathbb{P}_{X \sim P_b}\big[\frac{q_b}{p_b}(X) > t\big] + c\mathbb{P}_{X \sim P_b}\big[\frac{q_b}{p_b} = t\big]$ is the type 1 error of the log likelihood ratio test between $P_b$ and $Q_b$ with parameters $t$ and $c$, and $\alpha(t,c) = \sum_{b=1}^m w_b \alpha_b(t,c)$.*

In our case, all of the components correspond to continuous distributions and so we can ignore the $c$ parameter. This characterization is sufficient to compute and visualize the amplified tradeoff curves using our upper bound on the CDF of $B$ (Figure 1). For comparison with other DP variants, it can also be losslessly translated into a curve of $(\varepsilon, \delta)$ guarantees using Proposition 2.13 from Dong et al. (2019), from which we obtain the following corollary for our particular mixture distribution:

**Corollary 4.2.1.** *For all $\varepsilon > 0$, RW-AdaBatch satisfies $(\varepsilon, 1 - e^\varepsilon \alpha(\varepsilon) - \beta(\varepsilon))$-DP, with $\alpha$ and $\beta$ defined as in Lemma 4.2.2.*

**Algorithm 2** RW-Meta

**Require:** Noise scale $\eta$, learners $f_1, \ldots, f_m$
  Server samples $\tilde{G}^{(m)} \sim \mathcal{N}(0, \eta^2 I_m)$
  $\Sigma \leftarrow \eta^2 I_m$
  **for** $t = 1, \ldots, T$ **do**
    $\Sigma^* \leftarrow \Sigma - \frac{1}{m^2}(\vec{1}^T \Sigma \vec{1})\vec{1}\vec{1}^T$
    $\sigma^2 \leftarrow \max(2t, \lambda_{max}(\Sigma^*))$
    Server samples $y_t \sim \mathcal{N}(0, \sigma^2 I - \Sigma^*)$
    $j_t \leftarrow \arg\max_j (\tilde{G}^{(m)} + y_t)_j$
    **for** $i \in [m]$ **do**
      $x_{i,t} \leftarrow f_i(\tilde{g}_0, \ldots, \tilde{g}_{t-1}) \in \mathbb{R}^n$
    **end for**
    Clients send noisy vectors $\tilde{g}(D_{c,t})$ to server
    Server computes $\tilde{g}_t \sim \mathcal{N}(g_t, \eta^2 I_n)$ from the noisy vectors
    $X_t \leftarrow [x_{1,t}, \ldots, x_{m,t}]^T \in \mathbb{R}^{m \times n}$
    $\tilde{g}_t \leftarrow g_t + z_t$
    $\tilde{G}^{(m)} \leftarrow \tilde{G}^{(m)} + X_t \tilde{g}_t$
    $\Sigma \leftarrow \Sigma + \eta^2 X_t X_t^T$
  **end for**

**Complexity Analysis.** In the worst case, RW-AdaBatch requires us to find the root of a smooth monotonic function at each time step. Crucially, however, the size of the problem is constant and does not depend on $n$ or $T$, and so the asymptotic complexity of RW-AdaBatch is only $O(nT)$. We describe some of our practical techniques for improving the efficiency of this step in the appendix.

### 4.3 Dynamic Environments: Meta-learning

Recall that algorithms which minimize static regret, including RW-FTPL, often struggle in settings where the best action changes systematically over time. The RW-Meta algorithm we present in this section partially addresses this issue by showing how a key design pattern in many non-private, adaptive online learning algorithms can be efficiently made to satisfy DP. Specifically, RW-Meta uses a multilayer structure with many candidate algorithms (*learners*) along with a central *meta-learner* whose job is to select the best learner to use at each time step. The selected learner then determines our final action as a function of the (noisy) data seen so far. Our main technical innovation lies in the careful use of correlated noise and linearity, which allows us deploy much more powerful learners than prior work at no additional privacy cost.

**Utility Analysis.** We formalize our learners as a set of functions $f_1, \ldots, f_m : \mathbb{R}^{n,\infty} \to \mathcal{X}$. At time step $t$, each learner makes a prediction $x_{t,i} = f_i(\tilde{g}_1, \ldots, \tilde{g}_{t-1})$, and the meta-learner chooses $j_t \in [m]$. The meta-learner then receives the same gain as the chosen learner, i.e. $\langle x_{t,j_t}, g_t \rangle$. Our goal is to minimize the regret of the meta-learner with respect to the best single learner in hindsight.

The starting observation for our method is that the value $\langle x_{t,i}, \tilde{g}_t \rangle$ is an unbiased estimate for the gain of expert $i$ at time $t$. Specifically, let $X_t \in \mathbb{R}^{m \times n}$ be the matrix whose $i$th row is $x_{t,i}$. Then we have that $\sum_{s=1}^{t} X_t \tilde{g}_t \sim \mathcal{N}(G_t^{(m)}, \Sigma_t)$, where $G_t^{(m)} = \sum_{s=1}^{t} X_t g_t$ and $\Sigma_t = \eta^2 \sum_{s=1}^{t} X_t X_t^T$. So, like in RW-FTPL, our algorithm is able to maintain a Gaussian vector centered on the true gain of each learner, but with the inconvenient wrinkle that the covariance matrix is now data-dependent.

To dissolve this issue, we introduce a *decorrelation* step to the algorithm by defining a new matrix $\Sigma_t^* = \Sigma_t - \frac{1}{m^2}(\vec{1}^T \Sigma_t \vec{1})\vec{1}\vec{1}^T$. At each time step, we then sample a new Gaussian vector $y_t$ with mean zero and covariance matrix $\max(2t, \lambda_{max}(\Sigma_{t-1}^*))I - \Sigma_{t-1}^*$ and choose $j_t = \arg\max_j(\tilde{G}_{t-1}^{(m)} + y_t)_j$. Since our algorithm is invariant to additive noise with covariance $\vec{1}\vec{1}^T$, this is equivalent to the decision rule which simply adds Gaussian noise with covariance $\max(2t, \lambda_{max}(\Sigma_{t-1}^*))I$ at each step. In this way we can reduce to the scaled identity matrix covariance case, which is much easier to analyze. Using the convex analysis framework of Lee (2018), we arrive at a total regret bound of:

$$\left[\max\left(\sqrt{2},\ \eta\cdot\lambda_{max}\left(\frac{\Sigma_T^*}{\eta^2 T}\right)^{1/2}\right)+\sqrt{2}\right]\sqrt{2T\log m} \tag{3}$$

The best case scenario here is when all learners suggest different actions at each time step, in which case we enjoy an $O(\eta\sqrt{T\log m})$ regret bound, which is nearly optimal when $\eta$ is constant. The worst case occurs when the learners are divided into two cliques of size $m/2$, in which case $\lambda_{\max}(\Sigma_T^*/(\eta^2 T)) = m/2$ and we get a bound of $O(\eta\sqrt{Tm\log m})$. In both cases the regret bound is with respect to the best *learner*, which may be substantially better than the best *action*.

**Privacy Analysis.** The entire meta-learning algorithm accesses the data only through the $\tilde{g}_t$ vectors, and therefore satisfies local GDP with the same parameters as the base RW-FTPL algorithm.

**Complexity Analysis.** The RW-Meta algorithm requires $O(m^2 + mn)$ memory to store the predictions of the learners at each round as well as the $\Sigma$ matrix. Similarly, we require $O(m^2 + mn)$ operations per iteration to compute $\Sigma^*$ and $X_t\tilde{g}_t$. It is also necessary to choose a specific technique to compute $\lambda_{max}(\Sigma^*)$: in our implementation, we use the LOBPCG algorithm, which enjoys linear convergence and requires solving a $3 \times 3$ eigenproblem at each iteration (Knyazev, 2001). We can control the total number of iterations by warm-starting with the leading eigenvector of the previous iteration, which is guaranteed to be within $O(m)$ of the new maximum eigenvalue.

## 5    EXPERIMENTS

In this section, we empirically investigate the tightness of our analytic bounds on the privacy loss of RW-AdaBatch as well as the performance improvement of RW-Meta relative to prior work. The technical details of our experimental environment and implementation are described in the appendix, and we will release our data and scripts as open-source software prior to publication.

### 5.1    PRIVACY GUARANTEES OF RW-ADABATCH

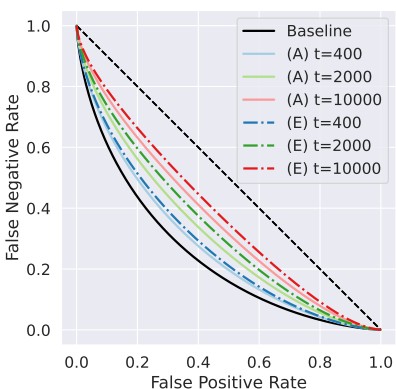 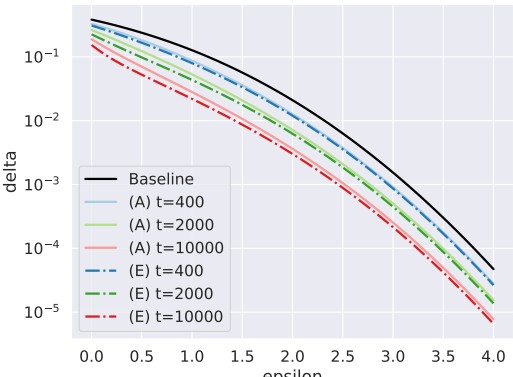

(a) Tradeoff curves representing the privacy amplification enjoyed by a point arriving at a given time step.

(b) Equivalent visualization in terms of $\varepsilon, \delta$ guarantees using Corollary 4.2.1.

Figure 1: Visualization of the privacy amplification provided by RW-AdaBatch with parameters $\mu = 1, n = 25, \Delta = \sqrt{n}, \alpha = 0.01$. The baseline corresponds to the $G_\mu$ tradeoff curve, solid lines correspond to the **A**nalytic upper bound from subsection 4.2, and dash-dotted lines correspond to **E**mpirical Monte Carlo simulations on zero-mean data with 1000 iterations.

On the basis of Theorem 4.2, we expect that the worst case for privacy occurs when all means are equal. To evaluate the privacy loss in this case, we simulate 1000 runs of RW-AdaBatch on a stream of all-zero data with $T = 10,000$. We then use the empirical PMF of containing batch sizes for each point to estimate its true tradeoff function. We plot these empirical tradeoff functions alongside those derived from our analytic bounds in Figure 1.

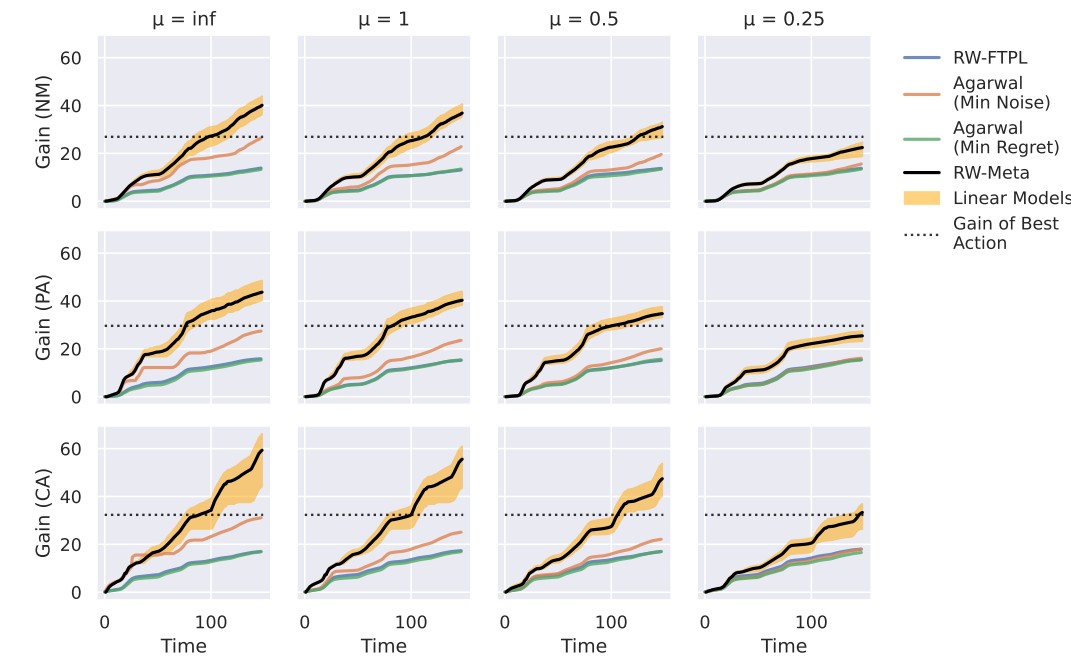

Figure 2: Results of empirical evaluation on COVID-19 hospitalization data, averaged over 100 iterations. The shaded orange regions enclose the maximum and minimum gain of the 12 rolling regression learners, and the dotted black lines represent the maximum cumulative COVID density of any single hospital in the given state. Static regret can be interpreted as the distance between a learner's total gain and the dotted line. Note that all plots share the same $x$ and $y$ axis.

Our results indicate that the analytic bounds are reasonably but not perfectly tight. The looseness primarily arises in the large $\delta$/moderate FPR regime. This is because in bounding the probability that the leader changes, we assume that the deterministic part of the gap will shrink at the fastest possible rate. In the oblivious adversary model, this is overly conservative because the adversary cannot predict in advance exactly when a new batch will begin. As a result, we underestimate the probability of selecting very large batch sizes. However, the low FPR behavior of the algorithm is primarily driven by the probability of seeing small batch sizes, where our bound is much tighter.

## 5.2 PERFORMANCE OF RW-META

**Methodology and Dataset**. While prior work on prediction with expert advice has largely relied on synthetic data alone (Erven et al., 2011; Korotin et al., 2020; Jun et al., 2017; Gao et al., 2024), we evaluate the performance of RW-Meta with real-world data from the COVID-19 pandemic. Specifically, we use data from the United States Department of Health and Human Services (DHHS, 2024) containing weekly reports from thousands of hospitals from mid-2020 through 2023. Our goal is to predict which hospitals will report the highest density of COVID patients each week.

Following Altieri et al. (2021), we consider models that first forecast the exact proportion of COVID cases in each hospital and then make a final prediction by taking the argmax of their forecast. The gain of a learner is the true proportion of COVID patients in the hospital they select. Our evaluation uses 13 learners consisting of rolling Gaussian regression algorithms with window sizes of 8/16/32/64 and weak/medium/strong regularization, as well as the basic RW-FTPL algorithm. We focus solely on Gaussian regression models because they are a popular choice for medical forecasting and their theoretical assumptions are a good match for the additive Gaussian noise we use to protect privacy. The specific range of hyperparameters we consider was chosen manually through exploratory analysis with a disjoint slice of the dataset, guided by the goal of maximizing variety while excluding non-functional or redundant learners. We evaluate our algorithm on data from three states (New Mexico, Pennsylvania, and California), which were chosen to cover a diverse range of population sizes while being geographically large enough to decorrelate different hospitals.

We remove any hospitals that did not appear to participate in the DHHS's data sharing program, defined as reporting fewer than 100 suspected cases over the 3 year period. This left us with data from 24, 146, and 293 hospitals in each state respectively, covering the 148 weeks between August 9th, 2020, and June 4th, 2023. We define adjacency to cover both whether an individual goes to the hospital and the specific hospital they choose, and compute the sensitivity of the gain vectors based on the minimum non-zero number of beds reported by any hospital at the given time step (minimum 4.3, maximum 14.6). We evaluate our algorithm under four different privacy levels: no privacy ($\mu = \infty$), low privacy ($\mu = 1$), medium privacy ($\mu = 0.5$) and high privacy ($\mu = 0.25$).

Finally, we compare RW-FTPL and RW-Meta against the state-of-the-art private FTPL algorithm of Agarwal & Singh (2017). This comparison must be heavily caveated because their algorithm only offers central DP guarantees and was not designed to operate in dynamic environments at all; we include these results not to demonstrate the superiority of one algorithm over the other, but to better contextualize the costs of satisfying local vs. central DP and the potential benefits of moving beyond the paradigm of data-independent experts. For fair comparison, we consider two possible strategies for setting the parameters of their algorithm: a greedy strategy that injects the minimum amount of noise required to still satisfy (central) $\mu$-GDP, and a cautious strategy that optimizes worst-case bounds on regret (and therefore might add more noise than required at the given privacy level).

**Results**. We repeat our evaluation 100 times and report average results, which are visualized in Figure 2 and summarized numerically in Table 1. Averages are reported alongside 95% confidence intervals based on the central limit theorem with Bonferroni correction. Across all settings, we find that RW-Meta significantly outperforms both RW-FTPL and the Min Regret variant. The Min Noise variant performs substantially better on this dataset despite weaker theoretical bounds, which is facilitated by the fact that, as a central DP algorithm, it can more easily control the amount of noise used. Nonetheless, it lags behind RW-Meta in all settings. This is because RW-Meta is able to consistently achieve *negative* static regret at low and moderate privacy levels, which is essentially impossible to match using the data-independent experts that are assumed in prior work. Meanwhile, when privacy is high, binary tree aggregation is able to offer only modest improvements in noise scale at the time horizon we consider. This is why why we see all three static experts algorithms clustering together when $\mu = 0.25$.

Separately, we find that RW-Meta performs around 90% as well as the best linear model in each setting on average. Which exact linear model performs best varies considerably across settings, however. For instance, the learner with window size 8 and strong regularization is the best performing learner in the high privacy setting for the Pennsylvania dataset, but only middle of the pack for New Mexico (where the best learner instead has a window size of 64). Across different privacy levels the variation is even larger, with the best learners in the low/no privacy settings performing the worst in the high privacy setting. This heterogeneity highlights the appeal of using meta-learning, as it eliminates the need to risk committing to a single set of parameters in advance.

## 6 CONCLUSION

We have presented two algorithms for the fundamental problem of prediction with expert advice, both of which satisfy local DP. In static environments, our RW-AdaBatch algorithm is a costless upgrade over the classical algorithm we build off of, improving the privacy of its outputs with provably insignificant impact on utility. In dynamic environments, our RW-Meta algorithm uses a novel, privacy-preserving variant of a classic online learning technique to quickly adapt to shifts in the input data, and we show that the resulting performance gains can be won at no additional privacy cost. Our analysis is supported by robust empirical evaluation, demonstrating that our algorithms are practical and can achieve high performance in a real-world, privacy-critical prediction task.

## REFERENCES

Jacob Abernethy, Chansoo Lee, Abhinav Sinha, and Ambuj Tewari. Online linear optimization via smoothing. In *Conference on learning theory*, pp. 807–823. PMLR, 2014.

Table 1: Average gain of learners

| Privacy Level | RW-Meta | Agarwal (Min Noise) | Agarwal (Min Regret) | Best Linear Model |
|---|---|---|---|---|
| New Mexico | | | | |
| $\mu = \infty$ | $40.14 \pm 0.30$ | $26.48$ | $13.38 \pm 0.38$ | $43.91$ |
| $\mu = 1$ | $36.87 \pm 0.36$ | $22.83 \pm 0.37$ | $13.41 \pm 0.39$ | $40.69 \pm 0.26$ |
| $\mu = 0.5$ | $31.19 \pm 0.55$ | $19.56 \pm 0.42$ | $13.25 \pm 0.40$ | $34.14 \pm 0.49$ |
| $\mu = 0.25$ | $22.46 \pm 1.04$ | $15.58 \pm 0.43$ | $13.38 \pm 0.43$ | $25.85 \pm 1.26$ |
| Pennsylvania | | | | |
| $\mu = \infty$ | $43.70 \pm 0.41$ | $27.42$ | $15.51 \pm 0.37$ | $48.61$ |
| $\mu = 1$ | $40.32 \pm 0.50$ | $23.55 \pm 0.44$ | $15.38 \pm 0.37$ | $44.30 \pm 0.39$ |
| $\mu = 0.5$ | $34.71 \pm 0.68$ | $20.03 \pm 0.44$ | $15.64 \pm 0.40$ | $38.06 \pm 0.57$ |
| $\mu = 0.25$ | $25.44 \pm 0.87$ | $16.09 \pm 0.42$ | $15.35 \pm 0.38$ | $28.73 \pm 0.82$ |
| California | | | | |
| $\mu = \infty$ | $59.36 \pm 0.47$ | $31.12$ | $16.68 \pm 0.36$ | $66.23$ |
| $\mu = 1$ | $55.62 \pm 0.47$ | $25.09 \pm 0.51$ | $16.91 \pm 0.34$ | $61.16 \pm 0.29$ |
| $\mu = 0.5$ | $47.43 \pm 0.81$ | $22.14 \pm 0.48$ | $16.82 \pm 0.41$ | $54.21 \pm 0.56$ |
| $\mu = 0.25$ | $33.34 \pm 1.30$ | $17.93 \pm 0.41$ | $16.84 \pm 0.35$ | $38.43 \pm 1.30$ |

Dmitry Adamskiy, Wouter M Koolen, Alexey Chernov, and Vladimir Vovk. A closer look at adaptive regret. In *Algorithmic Learning Theory: 23rd International Conference, ALT 2012, Lyon, France, October 29-31, 2012. Proceedings 23*, pp. 290–304. Springer, 2012.

Robert J Adler and Jonathan E Taylor. Gaussian inequalities. *Random Fields and Geometry*, pp. 49–64, 2007.

Naman Agarwal and Karan Singh. The price of differential privacy for online learning. In *International Conference on Machine Learning*, pp. 32–40. PMLR, 2017.

Nicholas Altieri, Rebecca L Barter, James Duncan, Raaz Dwivedi, Karl Kumbier, Xiao Li, Robert Netzorg, Briton Park, Chandan Singh, Yan Shuo Tan, et al. Curating a covid-19 data repository and forecasting county-level death counts in the united states. 2021.

Jason Altschuler and Kunal Talwar. Online learning over a finite action set with limited switching. In *Conference On Learning Theory*, pp. 1569–1573. PMLR, 2018.

Theodore W Anderson. The integral of a symmetric unimodal function over a symmetric convex set and some probability inequalities. *Proceedings of the American Mathematical Society*, 6(2): 170–176, 1955.

Hilal Asi, Vitaly Feldman, Tomer Koren, and Kunal Talwar. Private online prediction from experts: Separations and faster rates. In *The Thirty Sixth Annual Conference on Learning Theory*, pp. 674–699. PMLR, 2023.

Achraf Azize and Debabrota Basu. Concentrated differential privacy for bandits. In *2024 IEEE Conference on Secure and Trustworthy Machine Learning (SaTML)*, pp. 78–109. IEEE, 2024.

David Blackwell. Comparison of experiments. In *Proceedings of the second Berkeley symposium on mathematical statistics and probability*, volume 2, pp. 93–103. University of California Press, 1951.

Jean Bolot, Nadia Fawaz, Shanmugavelayutham Muthukrishnan, Aleksandar Nikolov, and Nina Taft. Private decayed predicate sums on streams. In *Proceedings of the 16th International Conference on Database Theory*, pp. 284–295, 2013.

Nicolas Carlini, Jamie Hayes, Milad Nasr, Matthew Jagielski, Vikash Sehwag, Florian Tramer, Borja Balle, Daphne Ippolito, and Eric Wallace. Extracting training data from diffusion models. In *32nd USENIX Security Symposium (USENIX Security 23)*, pp. 5253–5270, 2023.

Nicolo Cesa-Bianchi and Gábor Lugosi. *Prediction, learning, and games*. Cambridge university press, 2006.

T-H Hubert Chan, Elaine Shi, and Dawn Song. Private and continual release of statistics. *ACM Transactions on Information and System Security (TISSEC)*, 14(3):1–24, 2011.

Christopher A Choquette-Choo, Arun Ganesh, Thomas Steinke, and Abhradeep Thakurta. Privacy amplification for matrix mechanisms. *arXiv preprint arXiv:2310.15526*, 2023.

Christopher A Choquette-Choo, Arun Ganesh, Ryan McKenna, H Brendan McMahan, John Rush, Abhradeep Guha Thakurta, and Zheng Xu. (amplified) banded matrix factorization: A unified approach to private training. *Advances in Neural Information Processing Systems*, 36, 2024.

Thomas L Dean and Mark S Boddy. An analysis of time-dependent planning. In *AAAI*, volume 88, pp. 49–54, 1988.

Luc Devroye, Gábor Lugosi, and Gergely Neu. Prediction by random-walk perturbation. In *Conference on Learning Theory*, pp. 460–473. PMLR, 2013.

DHHS. Covid-19 reported patient impact and hospital capacity by facility, 2024. URL https://healthdata.gov/Hospital/COVID-19-Reported-Patient-Impact-and-Hospital-Capa/anag-cw7u/about_data. Accessed: 2024-11-20.

Jinshuo Dong, Aaron Roth, and Weijie J Su. Gaussian differential privacy. *arXiv preprint arXiv:1905.02383*, 2019.

John C Duchi, Michael I Jordan, and Martin J Wainwright. Local privacy, data processing inequalities, and statistical minimax rates. *arXiv preprint arXiv:1302.3203*, 2013.

Cynthia Dwork, Frank McSherry, Kobbi Nissim, and Adam Smith. Calibrating noise to sensitivity in private data analysis. In *Theory of Cryptography: Third Theory of Cryptography Conference, TCC 2006, New York, NY, USA, March 4-7, 2006. Proceedings 3*, pp. 265–284. Springer, 2006.

Cynthia Dwork, Moni Naor, Toniann Pitassi, and Guy N Rothblum. Differential privacy under continual observation. In *Proceedings of the forty-second ACM symposium on Theory of computing*, pp. 715–724, 2010.

Cynthia Dwork, Aaron Roth, et al. The algorithmic foundations of differential privacy. *Foundations and Trends® in Theoretical Computer Science*, 9(3–4):211–407, 2014.

Alexander Edmonds, Aleksandar Nikolov, and Jonathan Ullman. The power of factorization mechanisms in local and central differential privacy, 2019. URL https://arxiv.org/abs/1911.08339.

Úlfar Erlingsson, Vitaly Feldman, Ilya Mironov, Ananth Raghunathan, Kunal Talwar, and Abhradeep Thakurta. Amplification by shuffling: From local to central differential privacy via anonymity. In *Proceedings of the Thirtieth Annual ACM-SIAM Symposium on Discrete Algorithms*, pp. 2468–2479. SIAM, 2019.

Tim Erven, Wouter M Koolen, Steven Rooij, and Peter Grünwald. Adaptive hedge. *Advances in Neural Information Processing Systems*, 24, 2011.

Fengyu Gao, Ruiquan Huang, and Jing Yang. Federated online prediction from experts with differential privacy: Separations and regret speed-ups. 2024. URL https://arxiv.org/pdf/2409.19092.

Sascha Geulen, Berthold Vöcking, and Melanie Winkler. Regret minimization for online buffering problems using the weighted majority algorithm. In *COLT*, pp. 132–143. Citeseer, 2010.

Awni Hannun, Brian Knott, Shubho Sengupta, and Laurens van der Maaten. Privacy-preserving multi-party contextual bandits. *arXiv preprint arXiv:1910.05299*, 2019.

Charles R. Harris, K. Jarrod Millman, Stéfan J. van der Walt, Ralf Gommers, Pauli Virtanen, David Cournapeau, Eric Wieser, Julian Taylor, Sebastian Berg, Nathaniel J. Smith, Robert Kern, Matti Picus, Stephan Hoyer, Marten H. van Kerkwijk, Matthew Brett, Allan Haldane, Jaime Fernández del Río, Mark Wiebe, Pearu Peterson, Pierre Gérard-Marchant, Kevin Sheppard, Tyler Reddy, Warren Weckesser, Hameer Abbasi, Christoph Gohlke, and Travis E. Oliphant. Array programming with NumPy. *Nature*, 585(7825):357–362, September 2020. doi: 10.1038/s41586-020-2649-2. URL https://doi.org/10.1038/s41586-020-2649-2.

Elad Hazan and Comandur Seshadhri. Efficient learning algorithms for changing environments. In *Proceedings of the 26th annual international conference on machine learning*, pp. 393–400, 2009.

Mark Herbster and Manfred K Warmuth. Tracking the best expert. *Machine learning*, 32(2):151–178, 1998.

Naoise Holohan and Stefano Braghin. Secure random sampling in differential privacy. In *Computer Security–ESORICS 2021: 26th European Symposium on Research in Computer Security, Darmstadt, Germany, October 4–8, 2021, Proceedings, Part II 26*, pp. 523–542. Springer, 2021.

Prateek Jain, Pravesh Kothari, and Abhradeep Thakurta. Differentially private online learning. In *Conference on Learning Theory*, pp. 24–1. JMLR Workshop and Conference Proceedings, 2012.

Kwang-Sung Jun, Francesco Orabona, Stephen Wright, and Rebecca Willett. Online learning for changing environments using coin betting. 2017.

Peter Kairouz, Brendan McMahan, Shuang Song, Om Thakkar, Abhradeep Thakurta, and Zheng Xu. Practical and private (deep) learning without sampling or shuffling. In *International Conference on Machine Learning*, pp. 5213–5225. PMLR, 2021.

Adam Kalai and Santosh Vempala. Efficient algorithms for online decision problems. *Journal of Computer and System Sciences*, 71(3):291–307, 2005.

Andrew V Knyazev. Toward the optimal preconditioned eigensolver: Locally optimal block preconditioned conjugate gradient method. *SIAM journal on scientific computing*, 23(2):517–541, 2001.

Alexander Korotin, Vladimir V'yugin, and Evgeny Burnaev. Adaptive hedging under delayed feedback. *Neurocomputing*, 397:356–368, 2020.

Chansoo Lee. *Analysis of perturbation techniques in online learning*. PhD thesis, 2018.

Haipeng Luo and Robert E Schapire. Achieving all with no parameters: Adanormalhedge. In *Conference on Learning Theory*, pp. 1286–1304. PMLR, 2015.

Satya N Majumdar, Arnab Pal, and Grégory Schehr. Extreme value statistics of correlated random variables: a pedagogical review. *Physics Reports*, 840:1–32, 2020.

Albert W Marshall and Ingram Olkin. Majorization in multivariate distributions. *The Annals of Statistics*, 2(6):1189–1200, 1974.

Frank D McSherry. Privacy integrated queries: an extensible platform for privacy-preserving data analysis. In *Proceedings of the 2009 ACM SIGMOD International Conference on Management of data*, pp. 19–30, 2009.

Ilya Mironov. On significance of the least significant bits for differential privacy. In *Proceedings of the 2012 ACM conference on Computer and communications security*, pp. 650–661, 2012.

The mpmath development team. *mpmath: a Python library for arbitrary-precision floating-point arithmetic (version 1.3.0)*, 2023. http://mpmath.org/.

Victor Perrier, Hassan Jameel Asghar, and Dali Kaafar. Private continual release of real-valued data streams. *arXiv preprint arXiv:1811.03197*, 2018.

Reza Shokri, Marco Stronati, Congzheng Song, and Vitaly Shmatikov. Membership inference attacks against machine learning models. In *2017 IEEE symposium on security and privacy (SP)*, pp. 3–18. IEEE, 2017.

Abhradeep Thakurta and Adam Smith. (nearly) optimal algorithms for private online learning in full-information and bandit settings. *Advances in Neural Information Processing Systems*, 26, 2013.

Pauli Virtanen, Ralf Gommers, Travis E. Oliphant, Matt Haberland, Tyler Reddy, David Courna-peau, Evgeni Burovski, Pearu Peterson, Warren Weckesser, Jonathan Bright, Stéfan J. van der Walt, Matthew Brett, Joshua Wilson, K. Jarrod Millman, Nikolay Mayorov, Andrew R. J. Nelson, Eric Jones, Robert Kern, Eric Larson, C J Carey, İlhan Polat, Yu Feng, Eric W. Moore, Jake VanderPlas, Denis Laxalde, Josef Perktold, Robert Cimrman, Ian Henriksen, E. A. Quintero, Charles R. Harris, Anne M. Archibald, Antônio H. Ribeiro, Fabian Pedregosa, Paul van Mul-bregt, and SciPy 1.0 Contributors. SciPy 1.0: Fundamental Algorithms for Scientific Computing in Python. *Nature Methods*, 17:261–272, 2020. doi: 10.1038/s41592-019-0686-2.

Chendi Wang, Buxin Su, Jiayuan Ye, Reza Shokri, and Weijie Su. Unified enhancement of privacy bounds for mixture mechanisms via $f$-differential privacy. *Advances in Neural Information Processing Systems*, 36, 2024.

Tianhao Wang, Joann Qiongna Chen, Zhikun Zhang, Dong Su, Yueqiang Cheng, Zhou Li, Ninghui Li, and Somesh Jha. Continuous release of data streams under both centralized and local differential privacy. In *Proceedings of the 2021 ACM SIGSAC Conference on Computer and Communications Security*, pp. 1237–1253, 2021.

Mengmeng Yang, Taolin Guo, Tianqing Zhu, Ivan Tjuawinata, Jun Zhao, and Kwok-Yan Lam. Local differential privacy and its applications: A comprehensive survey. *Computer Standards & Interfaces*, pp. 103827, 2023.

Martin Zinkevich. Online convex programming and generalized infinitesimal gradient ascent. In *Proceedings of the 20th international conference on machine learning (icml-03)*, pp. 928–936, 2003.

## A  Appendix

### A.1  Limitations and Future Work

#### A.1.1  Applicability of Local DP

The primary limitation of our work is that satisfying local DP sometimes requires adding unreasonably large amounts of noise, particularly when gain vectors are high-dimensional. To guarantee low regret, our approach requires gain vectors with sensitivity smaller than their literal dimension, or else a silo-like setting where several sensitive values can be averaged locally before being sent to the algorithm, but this may not always be achievable in practice.

#### A.1.2  Extension to More Complex Learning Problems

While the experts problem is flexible enough to represent many real world tasks, it is also interesting from a theoretical perspective because it is one of the most fundamental problems in online learning. This naturally raises the question of whether the methods developed in this work can be extended to more complicated learning problems.

With respect to metalearning, RW-Meta fundamentally requires a linear gain function to compute unbiased estimates for the gain of each learner. So, while it can be extended to online linear optimization problems beyond prediction with expert advice, it is not clear that the same method could be generalized to the larger class of online convex optimization. On the other hand, it may be possible to extend RW-Meta to partial information settings such as linear bandits where it is possible to derive unbiased estimates of (linear functions of) the expected reward at each time step.

Meanwhile, RW-AdaBatch relies on the fundamental property that the maximum of a linear function over the convex hull of a small number of extreme points is always attained at one of those points. It would therefore be difficult to extend it to settings like online linear optimization over the sphere where the action set lacks this structure. On the other hand, the same property holds for maximums of convex functions over convex hulls, and so it is possible that the ideas behind RW-AdaBatch could be extended to some instances of online convex optimization.

## A.2  PROOF OF THEOREM 4.1

The proof of the theorem requires the following fundamental results about the extrema of Gaussian processes:

**Lemma A.0.1.** *Let* $z_1, \ldots, z_B \sim \mathcal{N}(0, \eta^2)$, *and let* $S_t = \sum_{i=1}^{t} z_i$. *Then:*

$$\max_t S_t \leq_{st} |S_B| \tag{4}$$

*Proof.* Instead of considering the discrete random walk, we consider the continuous analogue in one-dimensional Brownian motion over $[0, B]$ with scale $\eta$. The maximum value attained by the discrete random walk is at most as large as the maximum value attained by the Brownian motion. The latter is known to follow a half-normal distribution with scale $\eta\sqrt{B}$ (see e.g. Section 3 of Majumdar et al. (2020)), which is also the distribution of $|S_B|$. □

We additionally need the well-known Borell-TIS inequality, which states that the maximum of Gaussians concentrates closely around its expected value (Adler & Taylor, 2007):

**Lemma A.0.2** (Borell-TIS). *Let* $\{f_t\}$ *be a centered Gaussian process on* $T$. *Denote* $\|f\| = \sup_{t \in T} f_t$ *and* $\sigma_T = \sup_{t \in T} \sigma_t$. *Then for any* $u \geq 0$,

$$\mathbb{P}(\|f\| > \mathbb{E}[\|f\|] + u) \leq \exp\left(\frac{-u^2}{2\sigma_T^2}\right) \tag{5}$$

Finally, we have the following standard upper-bound on the expected maximum of Gaussians based on moment-generating functions:

**Lemma A.0.3.** *Let* $z_1, \ldots, z_n$ *be (not-necessarily independent) random variables such that* $z_i \sim \mathcal{N}(0, \sigma_i^2)$. *Let* $Z = \max_i z_i$. *Then:*

$$\mathbb{E}[Z] \leq \max_i \sigma_i \sqrt{2 \log n} \tag{6}$$

*Proof.* We have that:

$$\exp(t\mathbb{E}[Z]) \leq \mathbb{E}[e^{tZ}]$$
$$\leq \sum_{i \in [n]} \mathbb{E}[e^{tz_i}]$$
$$= \sum_{i \in [n]} \exp(\sigma_i^2 t^2/2)$$
$$\leq n \exp(\max_i \sigma_i^2 t^2/2)$$

Taking log of both sides and choosing $t$ to minimize the upper bound gives the desired result. □

Now: suppose that at a given time step, the gap of $\tilde{G}$ is $k$. Our goal is to bound the probability that our algorithm's output will change in the next $B$ time steps. Because the algorithm is maximizing a linear objective function, it will always output one of the vertices of the probability simplex. Without loss of generality, assume that the current time step is 0 and that the current leader has index 1, and write $S_{t,i} = \sum_{s=1}^{t} (\tilde{g}_t)_i$. Then the following is a necessary condition for our algorithm's prediction to change:

$$\max_{j>1} \max_{t \in [B]} S_{t,j} > k + \min_{t \in [B]} S_{t,1} \tag{7}$$

Lemma A.0.1 tells us that each maximum over $t$ is stochastically smaller than a half-normal distribution with scale $\eta\sqrt{B}$, and by symmetry the minimum on the right hand side is stochastically larger than a negative half-normal. We can write a half-normal random variable as the maximum of a Gaussian random variable and its negation, and so Lemma A.0.3 lets us bound the expectation of the left-hand side with $\eta\sqrt{2B\log(2n-2)}$. Denote this quantity as $E$. Then, assuming that $k > E$, we can use Lemma A.0.2 along with the independence of each coordinate to upper bound the probability of our event with:

$$\int_{-\infty}^{0} p(\min_t S_{t,1} = z)\mathbb{P}(\max_{t,j>1} S_{t,j} > k + z)\,dz$$

$$= 2\Phi\left(\frac{E-k}{\eta\sqrt{B}}\right)$$

$$+ 2\int_{0}^{k-E} \frac{1}{\eta\sqrt{B}}\varphi\left(\frac{E-k+u}{\eta\sqrt{B}}\right)\exp\left(\frac{-u^2}{2B\eta^2}\right)du$$

Here, the first term represents the probability that the current leader dips below the expected maximum value of the $n-1$ remaining actions, and the second term represents the probability that one of those $n-1$ actions manages to overtake the current leader regardless. Using the definition of the Gaussian PDF, we can rewrite the second term as:

$$\exp\left(\frac{-(E-k)^2}{4B\eta^2}\right)\int_{0}^{k-E} \frac{1}{\eta\sqrt{B}}\varphi\left(\frac{2u+(E-k)}{\eta\sqrt{2B}}\right)du$$

$$= \frac{1}{\sqrt{2}}\exp\left(\frac{-(E-k)^2}{4B\eta^2}\right)\int_{(E-k)/(\eta\sqrt{2B})}^{(k-E)/(\eta\sqrt{2B})}\varphi(y)\,dy$$

$$= \frac{1}{\sqrt{2}}\exp\left(\frac{-(E-k)^2}{4B\eta^2}\right)\left[\Phi\left(\frac{k-E}{\eta\sqrt{2B}}\right) - \Phi\left(\frac{E-k}{\eta\sqrt{2B}}\right)\right]$$

Write $\beta = (k-E)/(\eta\sqrt{2B})$. Then the entire upper bound can be simplified as:

$$2\Phi(-\sqrt{2}\beta) + 2\sqrt{\pi}\varphi(-\beta)\left[\Phi(\beta) - \Phi(-\beta)\right] \tag{8}$$

which proves the theorem. $\qquad\square$

## A.3 COMPUTEDELAY SUBROUTINE

==This appendix contains the explicit computation used by RW-AdaBatch to compute the size of the next batch:==

---

**Subroutine** `ComputeDelay`

**Require:** Noise scale $\eta$, gap $k$, dimension $n$, tolerance $\alpha$

$E \leftarrow \sqrt{\log(2n-2)}$

$U_1(B) := 2\Phi\left(\frac{B-k}{\eta\sqrt{B}} + \sqrt{2}E\right)$

$U_2(B) := 2\sqrt{\pi}\varphi\left(\frac{k-B}{\eta\sqrt{2B}} - E\right)$

$U_3(B) := \Phi\left(\frac{k-B}{\eta\sqrt{2B}} - E\right) - \Phi\left(\frac{B-k}{\eta\sqrt{2B}} + E\right)$

$\delta_t(B) := \alpha\sqrt{\frac{\log n}{t+B}}$

$B \leftarrow \texttt{FindRoot}\left(U_1(B) + U_2(B)U_3(B) - \delta_t(B)\right)$

**return** $\max(0, \texttt{Floor}(B))$

---

## A.4 PROOF OF THEOREM 4.2

The set $S_\varepsilon$ is invariant under permutations and shifts in the $\vec{1}$ direction, and so we can assume without loss of generality that $\mu_1 \geq \mu_2 \geq \ldots \geq \mu_n = 0$. It is easiest to begin by finding the measure of the

complement of our set, which is a union of disjoint sets where one element is the leader and the gap is greater than $\varepsilon$. This measure is given by:

$$f(\mu) = \gamma^n((S_\varepsilon - \mu)^C)$$

$$= \sum_{i=1}^{n} \int_{-\infty}^{\infty} \varphi(x - \mu_i) \prod_{j \neq i} \Phi(x - \varepsilon - \mu_j) \, dx$$

Our strategy is to show that the partial derivative of $f$ with respect to $\mu_1$ is non-negative. To do so, we can use the fact that each of our disjoint sets is invariant under shifts in the $\vec{1}$ direction, and so the gradient of each summand with respect to $\mu$ must be orthogonal to $\vec{1}$. So, we can write our partial derivative as:

$$\sum_{i=2}^{n} \int_{-\infty}^{\infty} \varphi(x - \mu_1)\varphi(x - \varepsilon - \mu_i) \prod_{j \neq 1, i} \Phi(x - \varepsilon - \mu_j) \, dx$$

$$- \sum_{i=2}^{n} \int_{-\infty}^{\infty} \varphi(x - \varepsilon - \mu_1)\varphi(x - \mu_i) \prod_{j \neq 1, i} \Phi(x - \varepsilon - \mu_j) \, dx$$

Combining like terms, this gives us:

$$\sum_{i=2}^{n} \int_{-\infty}^{\infty} \varphi(x - \mu_1)\varphi(x - \mu_i) \, \exp\left(\varepsilon x - \frac{1}{2}\varepsilon^2\right)$$

$$\cdot [\exp(-\varepsilon\mu_i) - \exp(-\varepsilon\mu_1)] \prod_{j \neq 1, i} \Phi(x - \varepsilon - \mu_j) \, dx$$

Then, since $\mu_1 \geq \mu_i$ and $\varepsilon > 0$, the term inside the brackets is non-negative. Therefore, as a sum of integrals of products of non-negative values, the whole expression is non-negative.

From here, the fact that $S_\varepsilon$ is closed under permutations means that, if $\mu_1 = \mu_i$ for some $i$, then they have the same partial derivative. So, we can explicitly construct a path from any arbitrary $\mu$ to the origin along which $f(\mu)$ is non-increasing: first reduce $\mu_1$ until it equals $\mu_2$, then reduce both until they equal $\mu_3$, and so on. This construction shows that the function is globally optimized at $\mu = 0$, as desired. $\qquad\square$

**Remark A.1.** *In the special case where $n = 2$, $S_\varepsilon$ is a convex set and the statement can be proved directly using Anderson's theorem (Anderson, 1955) or its extension by Marshall and Olkin for Schur-concave sets (Marshall & Olkin, 1974). When $n > 2$, however, the set is neither convex nor Schur-concave, necessitating the more explicit analysis here.*

## A.5 PROOF OF LEMMA 4.2.1

The proof follows from a theorem by Blackwell (1951), reproduced as Theorem 2.10 in Dong et al. (2019), which states that $\mathcal{T}(P, Q) \leq \mathcal{T}(P', Q')$ iff there exists a randomized algorithm proc such that $\mathsf{proc}(P) = P'$ and $\mathsf{proc}(Q) = Q'$. We will construct such an algorithm.

Suppose we receive the random input $(s_2, x) \sim Q \times \mathcal{N}(b, s_2^2)$ for some a priori unknown $b$. We first choose $s_1 = F_P^{-1}(F_Q(s_2))$. Since $s_2 \sim Q$, we have that $F_Q(s_2) \sim \mathsf{Unif}(0, 1)$ and therefore $s_1 \sim P$. Then, since $P \geq_{st} Q$, we have that $F_P^{-1} \geq F_Q^{-1}$ and therefore $s_1 \geq s_2$. So, we can sample $z \sim \mathcal{N}(0, s_1^2 - s_2^2)$ and release the tuple $(s_1, x + z) \sim P \times \mathcal{N}(b, s_1^2)$, which completes the proof. $\quad\square$

## A.6 PROOF OF COROLLARY 4.2.1

The proof relies on the Proposition 2.13 from Dong et al. (2019), which we reproduce here:

**Lemma A.0.4** (Primal to Dual)**.** *Let $f$ be a symmetric trade-off function. A mechanism is $f$-DP if and only if it is $(\varepsilon, \delta(\varepsilon))$-DP for all $\varepsilon \geq 0$ with $\delta(\varepsilon) = 1 + f^*(-e^\varepsilon)$, where:*

$$f^*(y) = \sup_{-\infty < x < \infty} yx - f(x) \qquad (9)$$

*is the convex conjugate of $f$.*

In our case, $f$ is only defined on $[0, 1]$, so we let $f(x) = \infty$ for $x \notin [0, 1]$ and the supremum is effectively taken over $0 \leq x \leq 1$.

To find the convex conjugate, we need to find the specific value $\alpha(t)$ that optimizes $y\alpha(t) - \beta(t)$ for a given $y$. To that end, define $h_y(\alpha) = y\alpha - \mathcal{T}(\mathcal{A}(D), \mathcal{A}(D'))(\alpha)$. Then we have that $f^*(y) = h_y(\alpha^*)$, where $\alpha^* = \inf\{\alpha \in [0, 1] : 0 \in \partial h_y(\alpha)\}$.

We have that $\alpha(t)$ is differentiable with respect to $t$, so we can compute that:

$$\frac{d}{dt} h_y(\alpha(t)) = \frac{d}{dt}\Big(y\alpha(t) - \sum_{b=1}^{m} w_b \beta_b(t)\Big)$$

$$= y \sum_{b=1}^{m} w_b \frac{d}{dt}\alpha_b(t) - \sum_{b=1}^{m} w_b \frac{d}{dt}\beta_b(t)$$

$$= y \sum_{b=1}^{m} w_b \Big(\frac{-1\sqrt{b}}{\mu}\varphi\Big(\frac{t\sqrt{b}}{\mu} + \frac{\mu}{2\sqrt{b}}\Big)\Big)$$

$$- \sum_{b=1}^{m} w_b \frac{\sqrt{b}}{\mu}\varphi\Big(\frac{t\sqrt{b}}{\mu} - \frac{\mu}{2\sqrt{b}}\Big)$$

Define:

$$z_b = \frac{t\sqrt{b}}{\mu} - \frac{\mu}{2\sqrt{b}}$$

Then the expression simplifies to:

$$\frac{-1}{\mu}\Big(\sum_{b=1}^{m} \sqrt{b}w_b y\varphi(z_b + \mu/\sqrt{b}) + \sum_{b=1}^{m} \varphi(z_b)\Big)$$

$$= \frac{-1}{\mu} \sum_{b=1}^{m} \sqrt{b}w_b\varphi(z_b)\Big[1 + y\exp\Big(\frac{-\mu}{\sqrt{b}}z_b - \frac{\mu^2}{2b}\Big)\Big]$$

$$= \frac{-1}{\mu} \sum_{b=1}^{m} \sqrt{b}w_b\varphi(z_b)\big(1 + y\exp(-t)\big)$$

Setting this equal to 0 and using the fact that $\varphi, w, b > 0$, we obtain that:

$$y\exp(-t) + 1 = 0$$
$$t = \ln(-y)$$

Denote this last quantity as $t_y$. From this it follows that $f^*(y) = h_y(\alpha(t_y))$, and therefore that our mechanism satisfies $\varepsilon, \delta(\varepsilon)$-DP for all $\varepsilon > 0$ and:

$$\delta(\varepsilon) = 1 - e^{\varepsilon}\alpha(t_{-e^{\varepsilon}}) - \beta(t_{-e^{\varepsilon}}) = 1 - e^{\varepsilon}\alpha(\varepsilon) - \beta(\varepsilon)$$

as desired. $\qquad\square$

## A.7 PROOF OF RW-META UTILITY BOUND

We formally analyze the regret of RW-Meta using the convex analysis framework of Lee (2018). Let $M(G) = \max_{x \in \mathcal{X}}\langle x, G\rangle$ be the baseline potential function, and for any set of distributions $\{\mathcal{D}_t\}$, define the smoothed potential function $\tilde{M}_t(G) = \mathbb{E}_{z \sim \mathcal{D}_t} M(G + z)$. Finding the expected regret of any randomized FTPL-style algorithm can be reduced to finding the regret of the deterministic algorithm which plays $\mathbb{E}_{z \sim \mathcal{D}_t}[\arg\max_{x \in \mathcal{X}} M(G+z)] = \nabla\tilde{M}_t$ at each time step $t$, which by Lemma

3.4 in Lee (2018) satisfies the following equality:

$$\text{Regret} = \sum_{t=1}^{T} \Big( \underbrace{\big( \tilde{M}_t(G_{t-1}) - \tilde{M}_{t-1}(G_{t-1}) \big)}_{\text{overestimation penalty}}$$

$$+ \underbrace{D_{\tilde{M}_t}(G_t, G_{t-1})}_{\text{divergence penalty}} \Big) + \underbrace{M(G_T) - \tilde{M}_T(G_T)}_{\text{underestimation penalty}}$$

Where $D_f(y, x) = f(y) - f(x) - \langle \nabla f(x), y - x \rangle$ is the *Bregman divergence*, which is a measure of how quickly the gradient of $f$ changes. Intuitively, the overestimation penalty represents the error that results when we fool ourselves into believing an action is better than it really is by adding too much noise, and the divergence penalty represents the error that comes from always playing one step behind the real objective function. Because we add zero-mean noise, the underestimation penalty is always negative by Jensen's inequality, and so to prove low regret it suffices to upper bound the first two terms.

For the overestimation penalty, we can use the convexity of $M$ to bound:

$$\tilde{M}_t(G_{t-1}) - \tilde{M}_{t-1}(G_{t-1})$$

$$= \mathbb{E}_{z \sim \mathcal{N}(0,1)}[M(G_{t-1} + \Sigma_t^{1/2} z) - M(G_{t-1} + \Sigma_{t-1}^{1/2} z)]$$

$$\leq \mathbb{E}_{z \sim \mathcal{N}(0,1)}[M((\Sigma_t^{1/2} - \Sigma_{t-1}^{1/2}) z)]$$

From which it follows by telescoping that the entire sum is upper bounded by:

$$\mathbb{E}_{z \sim \mathcal{N}(0,1)}[\Sigma_T^{1/2} z]$$

$$\leq \sqrt{2 \log(m) \max(\eta^2 T, \lambda_{max}(\Sigma_T^*))}$$

Where the last step follows from Lemma A.0.3. For the divergence penalty, we use Lemma 3.14 in Lee (2018) to bound:

$$D_{\tilde{M}_t}(G_t, G_{t-1}) \leq \frac{\sqrt{2 \log m}}{\eta_t}$$

From which it follows that the sum can be upper bounded by $\sqrt{2 \log m} \cdot \sum_{t=1}^{T} \frac{1}{\sqrt{2t}} \leq 2\sqrt{T \log m}$, giving us our final regret bound of:

$$\left[ \max \left( \sqrt{2}, \ \eta \cdot \lambda_{max} \left( \frac{\Sigma_T^*}{\eta^2 T} \right)^{1/2} \right) + \sqrt{2} \right] \sqrt{2T \log m} \tag{10}$$

## A.8 IMPORTANCE OF DECORRELATION IN RW-META

We know that the vectors $X_t \tilde{g}_t$ are unbiased Gaussian estimates of the true gain of each learner with covariance matrix $X_t X_t^T$. A natural question is whether it is possible to use these vectors directly in a FTPL-style algorithm and achieve low regret. In other words, is the decorrelation step in RW-Meta actually needed? There is reason to suppose that the induced correlation could actually be beneficial in certain contexts — for instance, if two learners always made the same prediction in all cases, then it would ensure that the meta-learner isn't 'distracted' by the redundancy. In this appendix, we describe some representative challenges that we encountered in attempting to prove non-trivial regret bounds for this computationally-simpler version of the algorithm.

The first difficulty is that to bound the overestimation penalty, we need to bound a sum of the form:

$$\sum_{t=1}^{T} \tilde{M}_t(G_{t-1}) - \tilde{M}_{t-1}(G_{t-1}) \leq \sum_{t=1}^{T} \mathbb{E}_{z \sim \mathcal{N}(0,1)}[M((\Sigma_t^{1/2} - \Sigma_{t-1}^{1/2}) z)] \tag{11}$$

The right-hand-side is the expected maximum of a zero-mean Gaussian, and so we can use Lemma A.0.3 to upper bound it in terms of the maximum diagonal entry of the matrix $(\Sigma_t^{1/2} - \Sigma_{t-1}^{1/2})^2$. In the case where each $\Sigma_t = \eta_t^2 \Sigma$ for some fixed $\Sigma$ and non-decreasing sequence $\eta_1, \ldots, \eta_T$ (which includes the base RW-FTPL algorithm as a special case with $\Sigma = I$), this gives us a telescoping series and we can bound the overestimation penalty as a whole with $\eta_T \sqrt{2 \log m \max_i \Sigma_{i,i}}$. Without constraints on the $\Sigma_t$, however, the telescoping behavior disappears and it is possible for the sum of maximal diagonal entries to be significantly larger than the maximal diagonal entry of the sum. We could force the series to telescope by upper-bounding the maximum diagonal entry of each matrix with its trace, but this is loose by a factor of $\sqrt{m}$ in the worst-case.

Continuing on, the divergence penalty describes how quickly the gradient of $\tilde{M}$ changes, and so the standard method for bounding it uses the Hessian $\nabla^2 \tilde{M}(G)$. Specifically, by Lemma 3.14 in Lee (2018), if there exists a constant $\beta$ such that $\text{Tr}(\nabla^2 \tilde{M}(G)) \leq \beta/\eta$ for all $G$, then:

$$D_{\tilde{M}}(G, g + G) \leq \beta \|g\|_\infty^2 / \eta \tag{12}$$

If our noise is drawn from a Gaussian with covariance $\eta^2 I$ then this term can be upper bounded neatly by $\frac{1}{\eta}\sqrt{2 \log m}$ (Lee, 2018). With arbitrary covariance matrices, however, the trace may end up being lower bounded by a constant (e.g. if the rows of the $X_t$ matrices all live in some $m - 1$ dimensional subspace), and the resulting bound isn't even sublinear! While we were able to derive stronger bounds with more detailed probabilistic analysis, the central problem is that the correlations cause the algorithm's behavior to be very *unstable*. Even late in the input, a previously-unseen pattern of agreement could dramatically change the algorithm's predictions. Based on this intuition and the central importance of stability in the design of low-regret algorithms, we believe that the decorrelation step in RW-Meta is genuinely necessary to guarantee strong worst-case performance and not merely an artifact of our analysis.

## A.9 Experimental Methodology

All experiments are carried out on a laptop running Ubuntu 22.04 with an intel i5-1135G7 CPU and 16GB of RAM. We implement our algorithms using Python 3.10, numpy 2.0.0 (Harris et al., 2020), scipy 1.14.0 (Virtanen et al., 2020), and mpmath 1.3.0 (mpmath development team, 2023).

Although RW-FTPL is very straightforward, Both RW-Meta and RW-AdaBatch include non-trivial computations as subroutines which must be implemented efficiently. We briefly describe our implementation choices here.

In the case of RW-Meta, we use the LOBPCG algorithm as implemented in scipy to find the leading eigenvalue of $\Sigma_t^*$ at each iteration, with the leading eigenvector from the previous round as an initial guess. We find that at the problem sizes we consider, this step is not a significant bottleneck (taking about 2ms per iteration), and we therefore do not pursue any further optimizations.

In the case of RW-AdaBatch, we take advantage of the fact that the bound from Theorem 4.1 is a monotonic function of $\beta$ by pre-computing many input/output pairs. To avoid issues with floating point precision when computing extreme values of the Gaussian PDF/CDF, we use the mpmath library for this task, which supports arbitrary precision arithmetic and numerical integration. We then interpolate the result with a monotonic cubic spline, which can be used to closely approximate the required value of $\beta$ necessary to achieve a given failure probability $\delta$. In practice, the discrete nature of the choice of batch size means that the small differences between the true inverse function and the interpolated approximation are insignificant, while the corresponding speed-up is dramatic.

Finally, prior work has shown that straightforward DP implementations are often vulnerable to attacks that take advantage of the idiosyncrasies of floating point numbers, leading to catastrophic privacy failures (Mironov, 2012). We therefore employ the secure random sampling method of Holohan & Braghin (2021) which renders these attacks computationally prohibitive.

