# OpenReview forum: "Private Learning Fast and Slow: Two Algorithms for Prediction with Expert Advice Under Local Differential Privacy"
_ICLR.cc/2025/Conference — Submitted to ICLR 2025_

### Official Review · Reviewer_qnzh · 2024-10-22

**Soundness:** 3
**Presentation:** 2
**Contribution:** 3
**Rating:** 6
**Confidence:** 3

**Summary:**

This paper addresses the problem of prediction with expert advice under local differential privacy, proposing two algorithms based on the classical "Prediction by random-walk perturbations" algorithm: (1) RW-AdaBatch, which enhances privacy by batching incoming data; and (2) RW-Meta, which adapts to data shifts by selecting from multiple candidate learners. The authors provide both theoretical analysis and empirical evaluation, demonstrating the advantages of the proposed algorithms.

**Strengths:**

1. Online prediction with expert advice is a fundamental problem in online learning. The investigation of this problem under local privacy constraints is crucial in both theory and practice due to the sensitive nature of machine learning tasks.
2. The authors substantiate their claims with rigorous theoretical analysis and empirical evaluation, demonstrating the high performance of the proposed algorithms.

**Weaknesses:**

1. The absence of a main theorem (like Theorem 2 in [1]) summarizing the regret of the proposed algorithms (in terms of $\varepsilon, \delta, n$ and $T$) limits the reader's ability to digest the results and identify the contributions effectively.
2. The paper lacks a comparison with existing private online learning algorithms. Though they are not designed for the setting considered here, a comparative analysis with existing private online learning algorithms could provide valuable insights into the privacy-utility tradeoff, particularly regarding the additional costs incurred when transitioning from central to local differential privacy.
3. It seems the proposed algorithms only improve the privacy/utility by a small constant factor, which may not be significant.

[1] Hilal Asi, Vitaly Feldman, Tomer Koren, and Kunal Talwar. Private online prediction from experts:
Separations and faster rates. In The Thirty Sixth Annual Conference on Learning Theory, pp.
674–699. PMLR, 2023.

**Questions:**

1. It was stated that the regret of RW-Meta in (3) is with respect to the best learner in the candidates (line 355). Could you clarify whether the regret is with respect to the gain of the best learner on non-noisy data (i.e., $g_1,\dots, g_T$) or on noisy data (i.e., $\tilde{g}_1,\dots,\tilde{g}_T$)?
2. The goal of RW-Meta is to choose a learner and follow its action at each time step. It seems this can be done by running RW-FTPL over the set of learners. Why not just run RW-FTPL over the set of learners?
3. Why did you compare RW-Meta to RW-FTPL in Table 1? I think it would be better to compare RW-Meta to the Linear Models instead of RW-FTPL. As shown in Figure 2, Linear Models outperform RW-FTPL a lot for all $\mu$'s. The performance of RW-Meta should largely rely on these Linear Models. Thus, listing the performance of the linear models would be more meaningful.
4. In line 161, does it mean that $v_{(k)}$ is the $k$-th smallest element? Since the gap $v_{(n)} - v_{(n-1)}$ seems to be the largest value minus the second largest value.
5. In line 195, is the distribution $n$-dimensional Gaussian?
6. What are the functions $\alpha$ and $\beta$ in Corollary 3.2.1?

---

> ### Author Response · Authors · 2024-11-24
>
> Thank you for your helpful comments and kind words about the importance of our problem! We respond to all of your questions and concerns one-by-one below:
>
> **W1:** We understand and sympathize with this frustration. One of the struggles in writing this paper has been that the main results we prove are very difficult to express in a clean, decontextualized way. We can say a lot about the privacy amplification of RW-AdaBatch both theoretically and empirically, but there’s no closed form expression for the exact tradeoff function that we can present as a standalone theorem. Similarly, the regret bound of RW-Meta is difficult to understand until the reader has the context to understand the meaning of the $\Sigma^*$ matrix. We absolutely understand that this makes things more difficult for the reader, and as part of our revision have tried to do a better job of highlighting our results in a simple and direct way at the beginning of each section whenever possible (e.g. “We show that the expected regret of RW-AdaBatch is at most $(1+\sqrt{2}\alpha)$ times greater than the expected regret of RW-FTPL”). To some extent, however, we fear that this weakness might be a necessary consequence of the kinds of results that we prove.
>
> **W2:** This is a valid observation, and so we have added in an explicit comparison with the FTPL algorithm of Agarwal and Singh 2017, which is the state of the art central-DP algorithm in the sorts of low-dimensional settings that we target. We agree that this comparison can help shed light on the cost of satisfying local DP, as well as the relative benefit of moving beyond the existing paradigm of data-independent experts used in prior work.
>
> **W3:** In the case of RW-AdaBatch, we wish to emphasize that the privacy gain we provide is very nearly free - there is only minimal computational overhead, and the impact on regret is provably insignificant. In that context, we would argue that even a moderate increase in privacy is noteworthy, particularly because a constant-factor improvement in privacy can normally only be achieved by a constant-factor degradation in utility.
>
> In the case of RW-Meta, we would push back against the characterization that we improve utility only by a small constant factor. In terms of absolute performance, we improve over RW-FTPL by a factor of more than 2x in most settings, and improve over the state of the art central DP algorithm by 50% or more. We feel that this constitutes a significant improvement, especially because in many settings we actually achieve negative static regret - given the fundamental limitations of data-independent experts, the gap in performance we observe is almost certainly not surmountable by future algorithmic improvements to static experts algorithms.
>
> **Q1:** The regret is with respect to the non-noisy data - this is defined slightly earlier at the beginning of the utility analysis section (line 352 of the updated version).
>
> **Q2:** This is a very good question which gets to the heart of our technical contribution. The key issue is that we need our entire system to satisfy DP, not just a single component. If we want to identify learners that have done well on the data so far, we need to expend privacy budget to do that. But at the same time, if we want non-trivial learners that can change their predictions based on the data so far, then we need to expend privacy budget on that too. If we treat these two components separately, then we’ll be forced to split the privacy budget between them and accept some combination of A) less accurate learners overall or B) less precise identification of the best learners.
>
> RW-Meta can basically be thought of as a technique to avoid having to make that choice. We expend 100% of our privacy budget on making the learners as accurate as possible, but still manage to select strong learners with reasonable accuracy by exploiting some useful properties of Gaussian noise as well as the linearity of our gain functions. This gives us higher overall utility at the same privacy cost compared to just using RW-FTPL over the learners directly.
>
> **Q3:** Thank you for the suggestion! We have revised Table 1 to include a column for the best-performing linear model in each setting, alongside the newly added central-DP algorithms.
>
> **Q4:** This is a typo, thank you for catching it!
>
> **Q5:** Yes, this has been clarified in the revision.
>
> **Q6:** These are the same $\alpha$ and $\beta$ functions as in the immediately preceding lemma (now 4.2.2), defined in terms of the specific mixture of Gaussians that we prove is a lower bound for RW-AdaBatch’s tradeoff function. We have added some extra text to clarify this.

---

> > ### Comment · Reviewer_qnzh · 2024-11-27
> >
> > Thank you for your efforts in addressing my questions. While the absence of a concise expression of the final result makes it challenging to compare with existing methods theoretically, I appreciate the extensive empirical evidence presented that demonstrates the advantages of the proposed algorithms.
> >
> > I still have some questions regarding my first inquiry. It seems my earlier question was unclear, because the $g_i$'s should be the gain instead of the data. To clarify: I am interested in whether the regret is measured with respect to the reward of *running* the best learner on non-noisy data, where the learner’s actions at each round are based on non-noisy data.
> >
> > If the answer is yes, could you briefly explain how this is achieved? According to line 363, the input to each learner is $\tilde{g}_i$, which could lead to actions that differ significantly from those in the non-noisy scenario. I am curious about how you relate the rewards in these two settings in the proof.

---

> > > ### Author Response · Authors · 2024-11-27
> > >
> > > Oh, yes, our apologies - we did misunderstand your earlier question. The regret is measured with respect to the reward of running the best learner on the noisy data. In other words, all of the learners are required to satisfy differential privacy, and our goal is to identify the best overall private learner (which could still be quite bad if our privacy constraints are severe).
> > >
> > > Visually, you can see this in Figure 2. The yellow envelope in the far left column $(\mu = \infty)$ represents the performance of the learners on the non-noisy data. As $\mu$ decreases, the performance of the learners decreases as well because they are forced to make their predictions using noisier data. Our main result for the utility of RW-Meta essentially says that the bold line can't be *too* far below the top of the yellow envelope in any given plot, but without specific assumptions about the learners we can't say anything about how adjacent plots in the same row will compare.
> > >
> > > Practically speaking, we try to control the gap between the noisy and non-noisy settings by using learners that are designed to be robust against additive Gaussian noise, which helps ensure that our regret bounds are still saying something meaningful about absolute performance even at high privacy levels.

---

> > > > ### Comment · Reviewer_qnzh · 2024-11-28
> > > >
> > > > Thank you for your explanation, which addressed my concern. I recommend including a brief discussion on this point in the final version.
> > > >
> > > > I have raised my score accordingly.

---

### Official Review · Reviewer_Misn · 2024-10-25

**Soundness:** 3
**Presentation:** 3
**Contribution:** 2
**Rating:** 5
**Confidence:** 4

**Summary:**

The paper studies the problem of distributed online prediction with expert advice under local DP constraints. They propose two algorithms RW-AdaBatch and RW-Meta. The paper provides a theoretical analysis of the proposed algorithms. Additionally, the paper provides experimental results using real-world data.

**Strengths:**

1. The paper is well-written. The background and related previous work are clearly explained. The algorithms (RW-AdaBatch and RW-Meta) are described in detail.


2. The paper includes experiments on real-world data from the COVID-19 pandemic.

**Weaknesses:**

1. The distributed setting is not fully explained. It is unclear how multiple players cooperate together in this distributed setting. Can they share their observations with others? Is there any communication between them?


2. Recent work on differentially private prediction with expert advice includes results for both pure $\varepsilon$-DP and approximate $(\varepsilon,\delta)$-DP [1,2], which are also cited in this paper. However, this paper only provides privacy guarantees for approximate DP. Can RW-AdaBatch or RW-Meta be extended to pure DP as well? If not, could you elaborate on the challenges involved?


[1] Asi, Hilal, et al. "Private online prediction from experts: Separations and faster rates." The Thirty Sixth Annual Conference on Learning Theory. PMLR, 2023.

[2] Asi, Hilal, et al. "Near-optimal algorithms for private online optimization in the realizable regime." International Conference on Machine Learning. PMLR, 2023.


3. The paper states, *"recent work has shown that very private algorithms can be forced to incur $O(T)$ regret by adaptive adversaries (Asi et al., 2023b). We therefore focus exclusively on oblivious adversaries in this work."* This statement is somewhat misleading and may benefit from clarification. Asi et al. (2023b) show linear regret for the pure DP case, while it is still possible to achieve sub-linear regret for approximate DP.


4. The paper does not provide a regret lower bound for the problem.

**Questions:**

How should noise scale $\eta$ be set to ensure that RW-AdaBatch or RW-Meta is $(\varepsilon,\delta)$-DP?

---

> ### Author Response · Authors · 2024-11-24
>
> Thank you for your positive comments and insightful questions! We address all of your questions one-by-one below:
>
> **W1:** We have reorganized the introductory sections of our paper to address this issue. In particular, we now have an explicit section on our Problem Setting which describes the specific communication model we have in mind. To answer the questions specifically, we do not consider any communication or coordination between clients: at each time step, they merely compute private functions of their local data and send them to the central server for post-processing into noisy gain vectors.
>
> **W2:** This is a really interesting question! Currently, our privacy analysis treats all of our algorithms as post-processings of Gaussian mechanisms, which don’t satisfy pure DP for any value of epsilon. So, if we wanted to provide pure DP guarantees, we would need to either change our algorithms or change something about our analysis.
>
> On the algorithmic side, the most obvious extension would be to replace our Gaussian noise with Laplacian. This would work reasonably well for RW-FTPL - the sum of independent Laplacians should converge to be approximately Gaussian fairly quickly, and so we would likely end up with a similar overall regret bound plus some error term. Conceptually, we could also imagine deriving a variant of RW-AdaBatch for this setting, but the proof techniques would look different (in particular, the analogue to continuous Brownian motion would no longer hold), and we expect that the overall level of privacy amplification would be weaker because Laplacian tails are heavier than Gaussian tails, necessitating more conservative batch sizes. Finally, we are skeptical that this approach would work at all for RW-Meta - our design and evaluation of that algorithm relies very heavily on the particular algebraic properties of Gaussians in a way that would be difficult to extend to any non-stable distribution.
>
> On the analysis side, the most plausible approach to our eyes would be to set aside local DP and focus on satisfying pure DP in the central setting. For a fixed time horizon, the set of possible outputs our algorithm can produce is finite (if exponentially large), and so it should certainly be true that our outputs satisfy pure DP for some finite epsilon. It’s less clear whether they could be made to satisfy pure DP with a small epsilon, however, or whether those guarantees could hold over arbitrary time horizons: there’s been some interesting recent work in this vein on the pure DP properties of Gaussian Noisy Max [1], but directly translating those results to our setting and using composition theorems would lead to an extremely high privacy cost very quickly. Getting a stronger guarantee would almost certainly require analyzing the entire sequence of outputs as a unit instead of reasoning timestep-by-timestep as we do currently.
>
> Overall, our feeling is that it is probably easiest to provide pure DP guarantees when designing algorithms in the FTRL regime, as in both recent papers by Asi et al. In the context of FTPL, there are a lot of reasons to want to use Gaussians, including stronger utility guarantees [2; Theorem 3.15], and that naturally pushes us towards approximate DP.
>
> [1] Lebensold, Jonathan, Doina Precup, and Borja Balle. "On the Privacy of Selection Mechanisms with Gaussian Noise." International Conference on Artificial Intelligence and Statistics. PMLR, 2024.
>
> [2] Lee, Chansoo. Analysis of perturbation techniques in online learning. Diss. 2018.
>
> **W3:** While Asi et al. (2023b) do prove lower bounds that are specific to pure DP, Theorem 9 and Theorem 11 of that paper show that approximate DP algorithms can also be forced to suffer linear regret by adaptive adversaries in some circumstances. We are open to being corrected on this, but at the moment we believe that the statement accurately reflects the results in that paper.

---

> > ### Author Response · Authors · 2024-11-24
> >
> > **W4:** We agree that a formal lower bound would be a very interesting contribution and a promising direction for future work. Heuristically, it seems very plausible that regret is lower bounded by $O(\sqrt{Tn\log n})$ under local DP. This is because one of the key quantities used when analyzing regret in FTPL algorithms is the expected $\ell_\infty$ error in the cumulative sum of gain vectors, which can be lower bounded by $\Omega(\sqrt{Tn \log n})$ using existing results for mean estimation under local DP[3; proposition 4]. Simultaneously, RW-FTPL is already able to achieve regret on that order, so no stronger lower bound is possible.  That said, we admit that we do not have a formal proof for this claim yet.
> >
> > We would also like to observe that non-trivial lower bounds on regret were not proved in the central setting until Asi et al. (2023b), over a decade after research on continual learning under differential privacy was initiated. Given the relative scarcity of prior work on the local setting, we hope that the absence of a lower bound in this paper is not seen as a fatal weakness.
> >
> > [3] Duchi, John C., Michael I. Jordan, and Martin J. Wainwright. "Local privacy, data processing inequalities, and statistical minimax rates." arXiv preprint arXiv:1302.3203 (2013).
> >
> > **Q1:** This can be done using the primal-dual characterization of $f$-DP, which we allude to in corollary 4.2.1. Concretely, a mechanism satisfies $\mu$-GDP if and only if it satisfies $(\varepsilon, \delta(\varepsilon))-DP$ for all $\varepsilon \geq 0$, where $\delta(\varepsilon) = \Phi(-\varepsilon/\mu + \mu/2) - e^\varepsilon \Phi(-\varepsilon/\mu - \mu/2)$ [4; Corollary 2.13]. So, given epsilon, we would solve numerically for the value of $\mu$ that gives our desired delta and then set the noise scale as $\eta = \Delta/\mu$.
> >
> > [4] Dong, Jinshuo, Aaron Roth, and Weijie J. Su. "Gaussian differential privacy." Journal of the Royal Statistical Society: Series B (Statistical Methodology) 84.1 (2022): 3-37.

---

> > > ### Comment · Reviewer_Misn · 2024-11-25
> > >
> > > Thank you for the detailed response. I have carefully read the rebuttal and the comments from other reviewers. I will maintain my original score.

---

### Official Review · Reviewer_2XWn · 2024-10-30

**Soundness:** 2
**Presentation:** 1
**Contribution:** 2
**Rating:** 3
**Confidence:** 2

**Summary:**

The paper considers online learning under local differential privacy (LDP), focusing more specifically on prediction with expert advice under full information setting in the oblivious adversary model. The authors propose 2 LDP algorithms and one that works with centralised DP, and analyze the utility and privacy of the proposed methods. The authors empirically test the proposed LDP methods on a real data example.

**Strengths:**

i) While the paper continues a well-established line of research on DP continual learning, it focuses on the LDP setting, which has fewer contributions.

ii) The writing is generally good, although there are some major caveats as described in rest of this review.

iv) Relaxing the assumption of having a trusted central party can be important.

**Weaknesses:**

i) The paper completely brushes over many details of the problem and of the proposed solutions, which makes it unacceptably cumbersome and error prone to read. For example, while the stated focus of the paper is a distributed setting with LDP, this is not easy to notice from the writing: the proposed algorithms and definitions do not explicitly mention any separate parties, nor communication steps or clearly state which party does what. This generally makes it bothersome to try and check how the proposed algorithms actually fit into the stated setting.

ii) Some of the claimed contributions seem inaccurate and somewhat overstated (see Questions below for details).

iii) The paper omits some empirical comparisons to existing baselines (see Questions below for details)

**Questions:**

### Update after discussion

I still recommend rejecting the paper as it currently stand: as I have mentioned in the comments, especially after the edits, the paper feels very unfinished to the point of being hard to understand. I therefore cannot recommend accepting the paper, as I am unsure if I have understood the presented work correctly based on the writing. I have lowered my confidence to better reflect this uncertainty.

### Comments before discussion

Questions and comments in decreasing order of importance:

1) Especially Sec2: currently, it is unnecessarily hard to try and figure out some basic assumptions you use. Please explicitly define what are neighbouring distributions and which neighbourhood relation you use, i.e., what do you actually try to protect with DP.
2) On the adaptive batching and resulting privacy: based on the abstract and stated contributions, I find it very surprising that the batching does not actually give any amplification in the LDP setting. Please rewrite the related sections to make this clearer from the beginning.
3) Related to the previous comment, as the adaptive batching algorithm assumes a trusted central party, its empirical performance should be compared to the existing methods that assume the same setting, e.g., Asi et al. 2023 (cited in the current paper).
4) Please explicitly consider your chosen setting when formulating the algorithms and the discussion.
5) As per the [note on arXiv](https://arxiv.org/abs/1802.02638), Ullman 2018 citen in the current paper has been withdrawn by the author. Please check the reference and update to the new version as instructed by the author.
6) Lines 313-14: why is $<x_{t,i}, \tilde g_{t}>$ unbiased estimate? Do you assume something specific on the learners?


## Minor comments etc. (no need to comment or acknowledge)

* Please fix typos: lines 121-22 extra dot after Jain et al.
* Lines 192-93: mention what is $G_{t-1}$ .
* Lines 308-09: should be learner $i$, not each learner?
* Alg 2 seems to be missing $\tilde g_0$.
* Lines 86-87: I would not understand what LDP means from this definition.

**Details Of Ethics Concerns:**

I have can see no specific ethical issues with paper.

---

> ### Author Response · Authors · 2024-11-24
>
> Thank you for your many helpful and constructive suggestions! We largely agree with all issues raised and have done our best to address them in the revision. We respond point-by-point below:
>
> **W1:** This is a very valid criticism, and we appreciate being pushed to be more exact! We have substantially revised several aspects of our presentation to make it more clear how our algorithms fit into our target setting. In particular, we have added a new Problem Setting section that addresses these questions directly and better connects our presentation of background material with our particular setting. As part of this, we also formally state the model of communication we consider and have incorporated this model into the presentation of our algorithms. We hope that these changes are able to address all of the issues you raise.
>
> **Q1:** We agree that these details were not explicit enough in our original paper. We have revised our presentation of differential privacy to be more specific to our particular setting. In particular, we now define adjacent datasets and DP explicitly in terms of the local statistics used to compute gain functions in our setting. We have also added an explicit statement in our evaluation section describing the precise adjacency definition we use for Covid data (i.e. two datasets are adjacent at a given time step if they differ only in whether an individual went to the hospital or in the hospital that individual went to).
>
> **Q2:** We have rewritten the relevant section of our contributions to make it more clear that the amplified privacy guarantees of RW-AdaBatch refer specifically to its outputs. More importantly, we have significantly rewritten and expanded the preamble to the privacy analysis of RW-AdaBatch to draw a more explicit connection between our work and the established literature on privacy amplification by shuffling, which studies the circumstances in which local DP algorithms can satisfy central DP with stronger privacy parameters. Our revisions should hopefully clarify that this is the specific form of privacy amplification we are interested in.
>
> **Q3:** We politely disagree with the characterization that RW-AdaBatch assumes a trusted central party. Although our analysis is focused on its amplified central-DP guarantee, the algorithm satisfies local DP with exactly the same parameters as RW-FTPL, and therefore does **not** require any additional trust assumptions.
>
> That said, reviewer qnzh has also commented on the lack of quantitative comparison with prior work in the central model, and so we have performed some additional experiments comparing our algorithms with the FTPL algorithm of Agarwal and Singh 2017. We did consider evaluating against Asi et al. 2023, but their algorithm is heavily tuned for high dimensional data and large time scales in a way that would make fair comparison very challenging. In contrast, Agarwal and Singh represent the current state of the art in the sorts of low-dimensional settings we target, and have the added benefit of naturally satisfying Gaussian DP.
>
> **Q4:** We have done our best to incorporate this suggestions as described above.
>
> **Q5:** Thank you for letting us know! We have updated the reference accordingly.
>
> **Q6:** We are not making any specific assumption here. The gain of learner $i$ is defined to be $\langle x_{t,i}, g_t \rangle$, and so the fact that $\langle x_{t,i}, \tilde{g}_t \rangle$ is an unbiased estimate follows from the fact that $\tilde{g}$ follows a multivariate normal distribution with mean $g_t$.

---

> > ### Comment · Reviewer_2XWn · 2024-11-25
> > **Comment on the changes**
> >
> > Thanks for the rebuttal, could you please post a draft with the changes from the original submission highlighted to make the evaluation easier?

---

> > > ### Author Response · Authors · 2024-11-25
> > >
> > > Absolutely! We have uploaded a draft with highlighting as suggested.

---

> > > > ### Comment · Reviewer_2XWn · 2024-11-29
> > > > **Final comments**
> > > >
> > > > Thanks for the rebuttal and for the effort in updating the draft and doing the highlights. Unfortunately I am not eager to change my score at this point:
> > > >
> > > > Despite the changes (and partly because of the new additions), several parts of the paper feel more like a draft or a workshop version; for example, the distributed setting still feels more like an after though (e.g. there is no explicit threat model, many specifics are simply not mentioned), things are often stated in a very general and non-specific way which makes it hard to pinpoint what exactly happens
> > > > (e.g. Alg 1 lines 223-24 server computes $\tilde g_t \sim \mathcal N(g_t, \eta^2I_n )$: how is this calculated, is this some unspecified general function $g_t=(f(\tilde g(D_{1,t}),\dots,\tilde g(D_{?,t})))$ as on lines 160-161, or,  writing $S_t$ for the set of clients chosen at step $t$ for updating, should this be $\sum_{c \in S_t} \tilde g(D_{c,t})$, which could make sense comparing to lines 234-35 but is not written out anywhere, or something else?), and some things are simply stated in a confusing manner (e.g. in the Problem Setting, lines 168-172 very much give the impression that up to the experiments in Sec 5, everything is based on 1 single client communicating at any given time step, while the algorithms have unspecified number of clients in plural sending updates at each time step).
> > > >
> > > > Also as a minor addition, for ease of reading, it might be helpful to add a table with all the notations somewhere.
> > > >
> > > > I would encourage the authors to spend some time and really focus on writing the paper clearly, as the contribution otherwise, as far as I can judge from the current version, seems nice. I have lowered my confidence score to better reflect the uncertainty I currently have about the paper.

---

> > > > > ### Author Response · Authors · 2024-12-02
> > > > >
> > > > > Lines 223-24 do indeed refer to an unspecified general function as described in the Problem Setting section. Our results require that the server is able to derive a Gaussian approximation of the true gain vector, but do not change based on how exactly that gets implemented in any given setting. We did not want to limit our focus to a single specific implementation in a way that would understate the generality of our method, which is why we have tried to state things in a less specific way (e.g. by simply saying that 'the server computes $\tilde{g}_t$' in our pseudocode and describing in Section 3.4 the formal requirements this computation needs to satisfy).
> > > > >
> > > > > We do appreciate that abstraction can make things less approachable. We actually meant for the example on line 168 to help with that issue by providing a simple, concrete instantiation to help readers think through our results, in the same way that a text might encourage the reader to think about $\mathbb{R}^2$ while proving a theorem about vector spaces generally. We didn't mean to imply that our results only apply in that specific instance, particularly since Section 5 defines gain vectors in a different way entirely (as you observe).
> > > > >
> > > > > In any case, we are grateful for the decision to reduce the confidence of your review even if the score did not change, and would like to thank you again for a very helpful and constructive discussion!

---

### Official Review · Reviewer_kNsQ · 2024-10-31

**Soundness:** 3
**Presentation:** 3
**Contribution:** 3
**Rating:** 6
**Confidence:** 3

**Summary:**

The paper introduces two algorithms, RW-AdaBatch and RW-Meta, for prediction with expert advice under the constraints of local differential privacy (LDP). The primary objective is to enable prediction in the LDP setting. RW-AdaBatch is designed for static environments and enhances privacy by adaptively batching data points, while RW-Meta uses meta-learning to improve predictions in dynamic environments. The paper validates these algorithms through theoretical analysis and empirical testing on COVID-19 hospitalization data, showing significant improvements in prediction accuracy under realistic privacy constraints.

**Strengths:**

- The paper is well-motivated, addressing a practical and novel problem. It introduces a classical method for solving privacy-preserving problems and proposes variations to address specific challenges.
- The writing is clear and well-structured.
- Both algorithms achieve near-optimal regret bounds (as claimed) and are supported by detailed privacy analyses.
- The experiment improvements seems significant.

**Weaknesses:**

- Can the authors provide specific cases where prediction with expert advice under LDP would be essential?
- The computational cost of RW-AdaBatch and RW-Meta appears substantial due to their batched nature and eigen value operation, potentially limiting scalability to very large datasets. Additionally, the datasets used are moderate in size. Can the authors provide a complexity analysis for computation and memory?
- I am not deeply familiar with prediction with expert advice, so a direct comparison of the regret achieved by these algorithms and previously established ones would be helpful. The bounds claimed to be near-optimal seem to compare with non-private lower bounds that do not involve any privacy parameters ($\varepsilon$, $\delta$, $\mu$). Can the authors comment on that? What are some LDP related lower bounds?

**Questions:**

- How to tune the hyperparameter $B$?

---

> ### Author Response · Authors · 2024-11-24
>
> Thank you for your insightful questions and positive comments about the strengths of our paper! We address the weaknesses and questions you list one-by-one below:
>
> **W1:** Beyond the medical forecasting task that we consider in our evaluation, we also believe our methods could be of interest for forecasting population movement and regional energy usage. Local differential privacy is attractive in these settings because individual level records can be highly revealing, while prediction with expert advice is a natural model for any forecasting task because it’s relatively easy to observe aggregate behavior after the fact. We have altered the description of our contributions in section 1.1 to highlight these other domains.
>
> Separately, because prediction with expert advice is one of the most fundamental problems in online learning, we believe it is a natural starting point to explore new methods or proof techniques for online learning generally. We have included some discussion along these lines in Appendix A.1.
>
> **W2:** We have added a section on computational complexity for both RW-AdaBatch and RW-Meta. While both algorithms involve non-trivial computations as subroutines, these computations don’t necessarily become more complex as the dataset grows in size. For instance, RW-AdaBatch only ever needs to find a root of a one-dimensional function regardless of how large $n$ and $T$ are. In fact, it should generally become computationally cheaper as $T$ increases because batch sizes generally increase over time, and no meaningful computation has to be performed in the middle of a batch.
>
> Meanwhile, RW-Meta only needs to find the eigenvector corresponding to the maximum eigenvalue rather than the whole eigensystem, which is substantially cheaper. Asymptotically speaking, the biggest challenge for scalability is that RW-Meta requires $O(m^2 + mn)$ memory and computation simply to compute and store the matrices it uses. However, the fact that the regret of RW-Meta with respect to the best learner is $O(\sqrt{m})$ in the worst case is already a good reason to avoid taking $m$ too large. In our own evaluation, we found that by far the biggest computational bottleneck came from constantly re-fitting our rolling regression models, while the metalearning itself took only a few milliseconds per iteration.
>
> **W3:** This is a very good question, and we have revised our presentation of our utility bounds for RW-FTPL and RW-Meta to make this more explicit in the paper itself. In both cases, the regret bounds are expressed in terms of the noise scale $\eta$ which implicitly depends on both our privacy parameter $\mu$ and the sensitivity $\Delta$. In some contexts (including our Covid evaluation), $\Delta$ is a fixed constant that doesn’t depend on $n$, and so we do in fact recover regret bounds that are within a multiplicative factor (based on $\mu$) of the non-private baseline. In the worst case, however, $\Delta$ can be as large as $\sqrt{n}$ and we get a regret bound of $O(\sqrt{Tn\log n})$. This matches known lower bounds on expected $\ell_\infty$ error for mean estimation under local DP[1; Proposition 4].
>
> **Q1:** There may be some misunderstanding here: the batch size $B$ isn’t directly set as a hyperparameter. During runtime, RW-AdaBatch selects batch sizes dynamically based on the current gap between the top 2 experts as well as our maximum tolerance for errors, which we set through the hyperparameter $\alpha$. This computation gets repeated every time a batch ends and a new batch size must be chosen. We have added a new section to the appendix (A.3) presenting this in more explicit detail.
>
> In practice, we find that the algorithm is not particularly sensitive to the exact choice of $\alpha$  - with $\alpha=0.01$, we have never witnessed RW-AdaBatch suffer higher regret than RW-FTPL, and  setting $\alpha$ as high as 1 gives only very small improvements in privacy. We would therefore recommend just using $\alpha=0.01$ in most applications without any tuning.

---

> > ### Comment · Reviewer_kNsQ · 2024-11-25
> >
> > Thank you for clarifying these points. Like the other reviewers, I still find the results somewhat hard to digest, although I believe the author has made their best effort. I will maintain a positive review with moderate confidence.

---

### Author Response · Authors · 2024-11-24
**Paper Revision**

Dear Reviewers and AC,

We appreciate your many thoughtful and insightful comments! We have uploaded a revised draft of our paper that should hopefully address all of the major issues raised. In addition to many smaller changes that we will describe in responses to individual reviews, we would like to draw the reviewers’ attention to three fairly significant changes from the original version:

- We have separated the previous section on ‘Background and Related Work’ into two: a ‘Related Work’ section which is essentially unchanged from the first draft, and a ‘Problem Setting’ section which is largely new. We intend for this second section to clarify several points that were ambiguous in our first draft and also to better connect our presentation of background concepts to the particular settings we consider.
- At the suggestion of reviewers 2XWn and  qnzh, we have performed additional experiments to compare the performance of our algorithms with the state-of-the-art FTPL algorithm of Agarwal and SIngh (2017), described in the Experiments section.
- We have expanded and rewritten our presentation of RW-AdaBatch to better clarify the specific nature of the privacy amplification we are studying.

To make room for these changes, we have moved the Limitations section to the appendix. We believe that our paper is much stronger now as a result of incorporating the reviewers’ suggestions, and greatly look forward to further discussion.

Warm regards,

The authors

---

> ### Author Response · Authors · 2024-11-25
> **Highlighting**
>
> At the suggestion of reviewer 2XWn, we have added highlighting to draw attention to our changes. This appears to have slightly perturbed the vertical spacing of the text, forcing Table 1 to be placed on page 11. The current text *can* fit in 10 pages without the highlighting, as demonstrated by the previous version, and so we hope this minor violation of the style guide during the rebuttal process won't be an issue.

---

### Meta-Review · Area_Chair_nvzS · 2024-12-20

**Metareview:**

The paper presents two algorithms for LDP learning with expert advice, and analyses their performance both theoretically and empirically.

In terms of strengths, the paper addresses a relevant problem and the methods are supported with theoretical as well as empirical evaluations.

In terms of weaknesses, individual reviewers raise a number of concerns ranging from unclear presentation of the distributed setting and lack of clear main theorem, general unpolished presentation and lack of lower bounds.

Overall, none of the reviewers strongly support the paper, while two support rejection even after the rebuttal and revision. In light of this, it is clear that the paper does not meet the bar for acceptance to ICLR.

**Additional Comments On Reviewer Discussion:**

All the reviewers reacted to the author rebuttal and noted that while some of their concerns were addressed, others remained. There was no further private discussion as the decision was clear.

---

### Decision · Program_Chairs · 2025-01-22

Reject